# Deciphering a hexameric protein complex with Angstrom optical resolution

**Hisham Mazal[1,2†], Franz-Ferdinand Wieser[1,2,3†], Vahid Sandoghdar[1,2,3]***

[1]Max Planck Institute for the Science of Light, Erlangen, Germany; [2]Max-Planck-Zentrum für Physik und Medizin, Erlangen, Germany; [3]Friedrich-Alexander University of Erlangen-Nürnberg, Erlangen, Germany

**Abstract** Cryogenic optical localization in three dimensions (COLD) was recently shown to resolve up to four binding sites on a single protein. However, because COLD relies on intensity fluctuations that result from the blinking behavior of fluorophores, it is limited to cases where individual emitters show different brightness. This significantly lowers the measurement yield. To extend the number of resolved sites as well as the measurement yield, we employ partial labeling and combine it with polarization encoding in order to identify single fluorophores during their stochastic blinking. We then use a particle classification scheme to identify and resolve heterogenous subsets and combine them to reconstruct the three-dimensional arrangement of large molecular complexes. We showcase this method (polarCOLD) by resolving the trimer arrangement of proliferating cell nuclear antigen (PCNA) and six different sites of the hexamer protein Caseinolytic Peptidase B (ClpB) of *Thermus thermophilus* in its quaternary structure, both with Angstrom resolution. The combination of polarCOLD and single-particle cryogenic electron microscopy (cryoEM) promises to provide crucial insight into intrinsic heterogeneities of biomolecular structures. Furthermore, our approach is fully compatible with fluorescent protein labeling and can, thus, be used in a wide range of studies in cell and membrane biology.

**\*For correspondence:**
vahid.sandoghdar@mpl.mpg.de

[†]These authors contributed equally to this work

## Editor's evaluation

This paper will be of interest to the structural biology community and people working on cryogenic fluorescence microscopy. This paper is a clear step forward in the use of single-molecule localization microscopy at Å resolution, thanks to low-temperature polarized super-resolution imaging and advanced data processing algorithms.

## Introduction

Proteins and their various assemblies are among the main constituents of all living systems and govern every aspect of cellular physiology in both healthy and disease states (*Nelson, 2017*; *Mavroidis et al., 2004*; *Schliwa and Woehlke, 2003*; *Alberts, 1998*). These biomolecular structures adopt sophisticated three-dimensional (3D) configurations during their multifaceted conformational changes. A full understanding of their spatial arrangements and associated heterogeneous configurations is crucial for elucidating their molecular mechanisms, helps guide the engineering of new proteins, and is a great asset for drug discovery (*Renaud et al., 2018*). Indeed, since the pioneering work of Perutz on protein crystals (*Fersht, 2008*), a variety of techniques such as X-ray crystallography (*Shi, 2014*) and nuclear magnetic resonance (NMR) spectroscopy (*Kanelis et al., 2001*) have been explored for gaining insight into protein structure and function. Advances in sample preparation, detector technology, and image processing based on single-particle analysis have also ushered in atomic resolution in cryogenic electron microscopy (cryoEM) studies of protein structure (*Nakane et al., 2020*;

*Kühlbrandt, 2014*). The inherent resolution of this method is highly desirable, but lack of specific labeling makes it challenging to identify small variations such as structural inhomogeneities. Fluorescence microscopy, on the other hand, draws its success from an exquisite specificity in labeling, but has traditionally suffered from a limited resolution.

The recent advent of super-resolution (SR) fluorescence microscopy has opened new avenues for studying subcellular organization and is on the way to become a workhorse for biological studies (*Lelek et al., 2021*; *Sahl et al., 2017*; *Weisenburger and Sandoghdar, 2015*). However, conventional SR microscopy performed at room temperature is still not considered as a contestant in the arena of structural biology, where Angstrom-level information about the molecular architecture of proteins and protein complexes is sought after. To push the limit of fluorescence microscopy, one can perform measurements under cryogenic conditions (*Böning et al., 2021*, *Hoffman et al., 2020*; *Dahlberg et al., 2020*; *Moser et al., 2019*; *Wang et al., 2019*; *Hulleman et al., 2018b*, *Xu et al., 2018*; *Weisenburger et al., 2017*; *Furubayashi et al., 2017*; *Li et al., 2015*; *Weisenburger et al., 2014*; *Weisenburger et al., 2013*). In addition to slowing down photochemistry, which allows each fluorophore to emit several orders of magnitude more photons than at room temperature (*Li et al., 2015*; *Weisenburger et al., 2014*; *Weisenburger et al., 2013*), a key advantage of cryogenic temperatures is in offering superior sample preservation and high stability for Angstrom-scale structural studies. In one implementation, cryogenic optical localization in 3D (COLD) was introduced, where the stochastic intensity blinking of organic dyes gave access to the positions of up to four labeling sites on a single protein (*Weisenburger et al., 2017*). We recently employed a more robust protocol to identify individual fluorophores by exploiting the polarization of the fluorescence light dictated by the fluorophore orientation (*Böning et al., 2021*). This latter method was validated by measuring single distances on one-dimensional DNA nanorulers (*Böning et al., 2021*). Control of the fluorescence signal via polarization modulation has also been shown to offer an alternative to random blinking (*Hulleman et al., 2018a*, *Hafi et al., 2014*). In our current study, we introduce polarCOLD, which exploits polarization encoding for resolving several fluorophores in 3D. Importantly, we show that the distances and arrangements of protein complexes can be determined by combining images recorded from under-sampled structures. To demonstrate this, we first resolve three fluorophores on a trimer protein complex with Angstrom resolution. Next, we use partial labeling, a supervised particle classification procedure to solve a complete hexameric protein arrangement. We discuss the limits of our methodology for resolving structures with a certain degree of disorder as well as its promise for combination with cryoEM.

## Results

### polarCOLD on a protein trimer

Proliferating cell nuclear antigen (PCNA) is a central functional unit in genome repair and replication (*Bruck and O'Donnell, 2001*). The structure of this complex protein was solved by X-ray crystallography (*Georgescu et al., 2008*) and more recently by cryoEM (*Madru et al., 2020*). These studies have shown that PCNA forms a stable homo-trimer with a pseudo-hexameric shape. The stability and simple configuration of its structure make PCNA a good model system for benchmarking our imaging methodology. To study human PCNA with polarCOLD, we first fully labeled it via a His-tag linker on the N-terminal side of each subunit of the protein, forming an equilateral triangle (see Materials and methods section and *Figure 1—figure supplement 1*). PCNA complexes were then embedded in a hydrophilic poly-vinyl alcohol (PVA) matrix at sub-nanomolar concentration and spin-coated on a mirror-enhanced substrate (see *Böning et al., 2021* for in-depth characterization of the mirror-enhanced substrate). The resulting density corresponds to fewer than one protein per µm$^2$ on average so that individual proteins can be easily identified in diffraction-limited imaging. The samples were immediately imaged in our custom-built microscope (see Materials and methods section).

*Figure 1a* shows a schematic of the imaging setup. A liquid helium cryostat houses the cold stage as well as a scannable microscope objective (*Weisenburger et al., 2017*; *Böning et al., 2021*). A polarizing beam splitter in the detection path allows us to determine the orientation of dipole-like emitters projected onto the angular interval θ ∈ [0°, 90°] in the imaging plane. The high photostability of the fluorophores at T=4 K allows us to collect on average 260 photons per frame (per 14 ms) from a single fluorophore with a total number of registered photons exceeding 10$^6$ after 50,000 frames

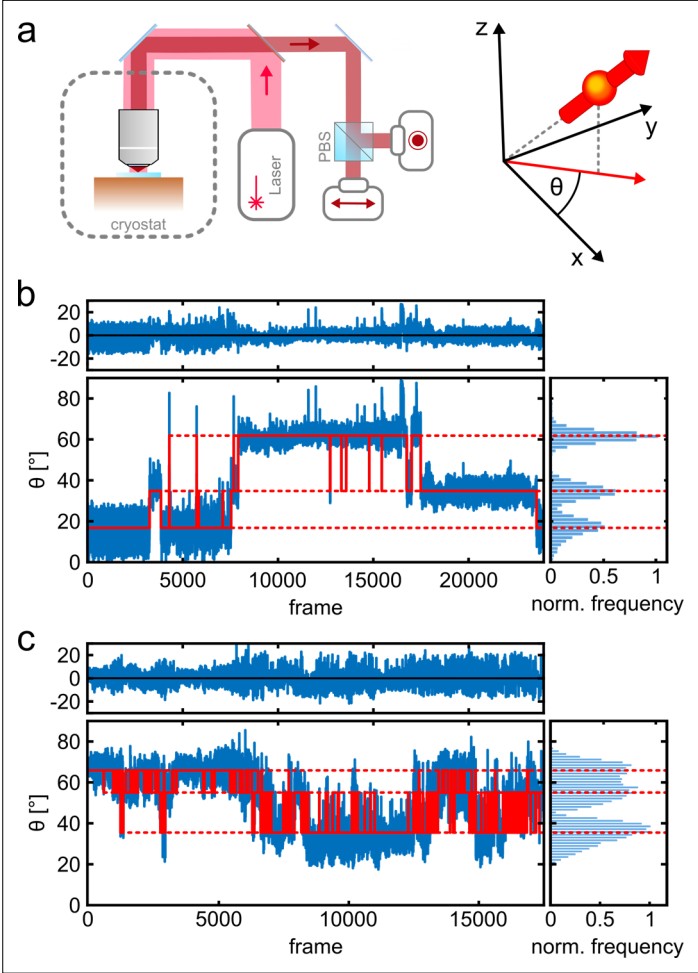

**Figure 1.** Photo-physics and co-localization of fluorophores at cryogenic temperatures. (**a**) Schematics of the cryogenic optical microscope. Polarization-resolved detection allows for direct measurement of the in-plane dipole moment of fluorophores. Here we use circularly polarized light from a laser at $\lambda$ =635 nm. A polarizing beam splitter in the detection path allows one to resolve the polarization state of each individual molecule. (**b, c**) Exemplary polarization time traces of two single proteins. (**b**) demonstrates a case of well-separated polarization states, whereas (**c**) displays a case with smaller separations between polarization states. Blue traces present the experimental polarization values for each frame, and the red lines show the polarization determined by the algorithm (*White et al., 2020*). Top panels shows the residuals of the fit. The blinking kinetics are exceptionally slow with on/off times in the range of seconds to minutes.

The online version of this article includes the following source data and figure supplement(s) for figure 1:

**Source data 1.** Exemplary polarization time traces of two single proteins for *Figure 1b-c*.

**Source data 2.** Overview of the recorded data and the experimental yield.

**Figure supplement 1.** Characterization and labeling of human PCNA protein.

**Figure supplement 1—source data 1.** Raw Native-page and fluorescent gel images.

**Figure supplement 2.** Photo-physics characterization of NTA-ATTO647N attached to human PCNA protein imaged at 4K.

**Figure supplement 3.** Example of complete intensity and polarization trace processing for fluorophore localization.

**Figure supplement 3—source data 1.** Raw data set used as an example for complete signal processing calculated from the same single molecule.

**Figure supplement 4.** Validation of polarization assignment.

**Figure supplement 5.** Polarization histogram of single fluorophore ATTO647N imaged at 4 K.

**Figure supplement 6.** Fluorophore identification using polarization-resolved imaging.

(*Figure 1—figure supplement 2*). *Figure 1b–c* (blue trace) displays two examples in which three different polarization states recur at various times. The long photo-blinking off-times (*Figure 1—figure supplement 2d*) allow one to identify each of the three fluorophores on a given protein complex separately (*Weisenburger et al., 2017*). More insight into polarization trace processing can be found in *Figure 1—figure supplement 3*. To resolve the polarization histograms in cases where they partially overlap (see, e.g., *Figure 1c*), we fit the data using an algorithm that combines unsupervised statistical learning tools with change-point detection in a model-independent manner (*White et al., 2020*). As illustrated by the red traces in *Figure 1b and c*, we can robustly identify the polarization states over time and hence assign the signal in each frame to a single fluorophore. We also verified the robustness of our assignment procedure by performing random assignment of frames, which resulted in single, unresolved spots (see Materials and methods section and *Figure 1—figure supplement 4*).

The number of fluorophores that can be simultaneously resolved depends on the blinking on-off dynamics (see Materials and methods section). Furthermore, the shot noise determines the angular resolution and, thus, the maximum number of resolvable polarization states per protein. This, in turn, directly affects the yield of resolved particles (see *Figure 1—figure supplements 5–6*). For example, in our current experiment, we used one polarization basis and projected all orientations to the limited space of $\theta \in [0°, 90°]$. By taking the experimental angular resolution of ca. 5° (*Figure 1—figure supplements 5–6*), we can theoretically expect to resolve 70% of the particles which contain 3 fluorophores, and roughly 15% of the particles which contain 6 fluorophores. Adding a second polarization basis at a tilt of 45°, would allow one to double the angular space (*Stallinga and Rieger, 2012*). Moreover, using polarized illumination (*Backer et al., 2016*; *Zhanghao et al., 2019*) and a full 3D characterization of the dipole moments (*Lieb et al., 2004*; *Mortensen et al., 2010*; *Hulleman et al., 2021*) would result in less overlap and thus enhanced capacity for identifying different polarization states. We remark, however, that even in our current scheme, one can improve the number of resolved polarization states by excluding the ambivalent cases, albeit at the expense of the overall yield (see *Figure 1—figure supplement 6* and *Figure 1—source data 2* for the statistics of this analysis).

Having identified the individual fluorophores, we generate super-resolved images by clustering the respective coordinates and taking their averages (see *Figure 2—figure supplements 1 and 2* for moving from traces to 2D resolved image). The top and bottom rows in *Figure 2a* display a selection of the measured and simulated 2D projection maps. To quantify their similarity, we computed a correlation score ranging from 0 to 1. We obtained 2D correlation scores of 0.92 or higher, representing nearly perfect agreement. To obtain a 3D model from our 2D localization maps, we use a single-particle reconstruction algorithm (*Dvornek et al., 2015*; *Weisenburger et al., 2017*; see Materials and methods section and *Figure 2—figure supplement 3* for complete data set). *Figure 2b* shows that the reconstructed fluorophore volumes (red spheroids) agree well with the crystal structure of the PCNA protein (PDB: 1AXC) containing three identical subunits in an equilateral triangle. The slight asymmetry and deviation from the actual crystal structure can in part be attributed to the uncertainty introduced by the dye linker, 6-histidine linker and possibly the restricted rotational mobility of the dye itself, resulting in a minor localization bias. Indeed, by taking the dye linker into account and calculating the accessible volume (*Kalinin et al., 2012*), we found that our 3D reconstructed volumes correlate very well (0.96 correlation score) with the simulated accessible volumes (see *Figure 2b*, *Animation 1*).

To evaluate the resolution of our 3D reconstructed volumes, we used the well-established method of Fourier shell correlation (FSC) (*van Heel and Schatz, 2005*). Here, we divide the 2D image data set into two randomly chosen groups and then determine their 3D reconstruction separately. Then we assess the cross-correlation (similarity) between the two 3D volumes in Fourier space as a function of spatial frequency. The overall resolution of a 3D reconstructed volume is thus obtained by finding the maximum spatial frequency corresponding to a correlation above a specific threshold value. Here, we used the half-bit criterion, which is a standard threshold curve used in single-particle cryoEM. As shown in *Figure 2c*, the intersection of this curve (red) with the FSC curve (blue) indicates at which spatial frequency we have collected a sufficient amount of information in order to interpret the 3D reconstructed volumes accurately (*van Heel and Schatz, 2005*). We find a remarkable resolution of 4.9 Å. We further quantified the size of the protein-dye conjugate via the pair-wise distances between the localized sites on each particle. The histogram in *Figure 2d* plots the distribution of the side lengths of the projected triangles and is well-described by a fit that considers the localization

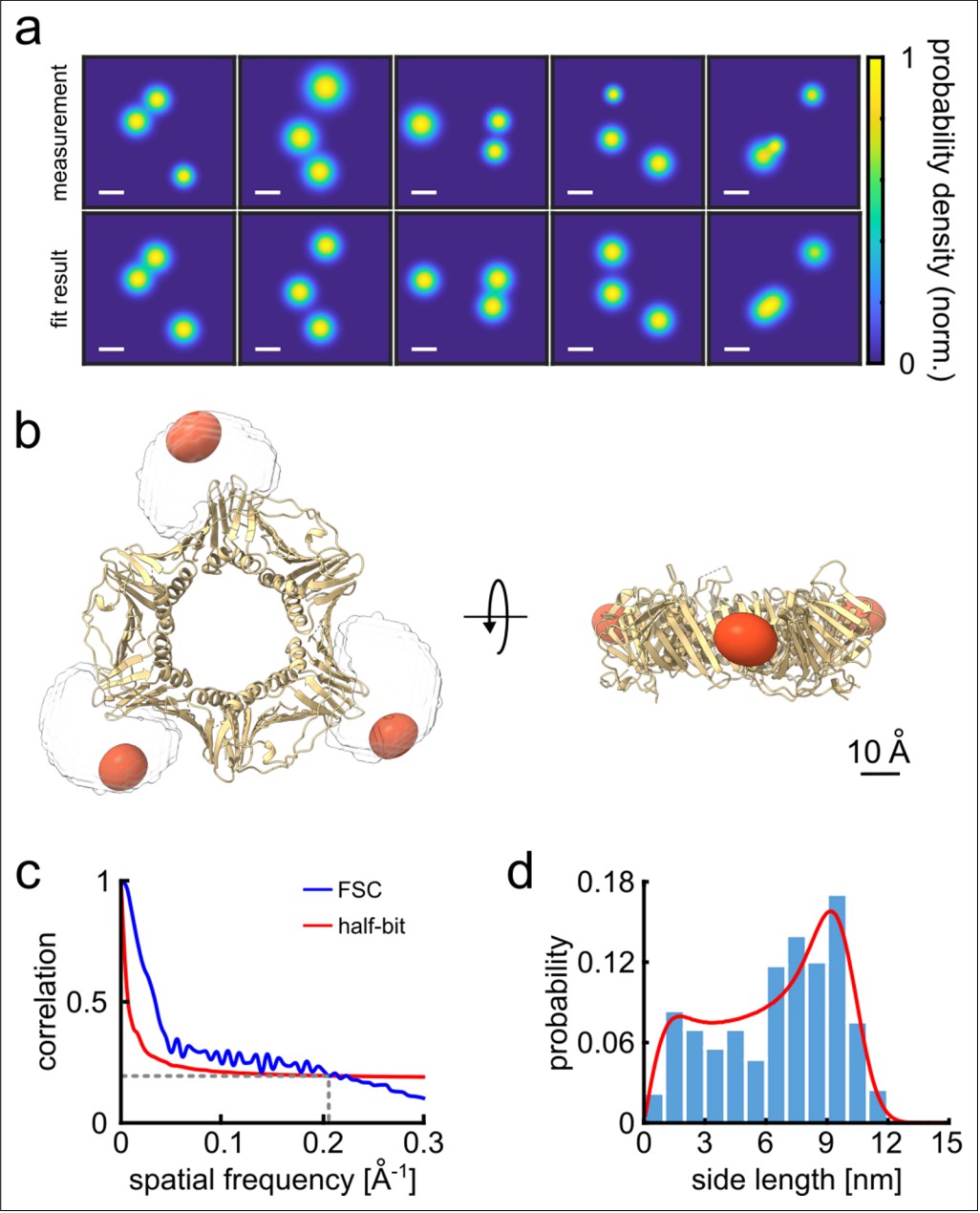

**Figure 2.** 3D reconstruction of the PCNA protein trimer. (**a**) Experimentally obtained super-resolved 2D images (top row) of single proteins and simulated images based on the crystal structure (bottom row). The color code represents the occupation probability determined by the localization precision for each fluorophore. The localization precision in the simulated data was normalized. Scale bar is 3 nm. (**b**) Overlay of the crystal structure of human PCNA with the reconstructed fluorophore volumes shown as red spheroids (see online Animation 1). The transparent white clouds represent the accessible volume of the dye linker attached to the N-terminal side of the protein, calculated using the parameter of ATTO647N as provided in *Kalinin et al., 2012*. By fitting the reconstructed 3D volumes obtained from polarCOLD into the theoretical accessible volumes of the dyes, we find a correlation score of 0.96, indicating a correct 3D reconstruction. (**c**) The Fourier shell correlation (FSC, blue curve) of the two half data sets gives a resolution of 4.9 Å based on the half-bit criterion (red curve). (**d**) Distribution of the projected side lengths (blue) obtained from the localized positions shown in (**b**). The model fit (red) takes the finite localization uncertainty and the random particle orientation into account, resulting in 9.9±0.6 nm. The error of the model fit was estimated from 200 fits. The reconstructed 3D volume was calculated from 119 particles.

The online version of this article includes the following source data and figure supplement(s) for figure 2:

**Source data 1.** Full dataset coordinates of the 2D images used for 3D reconstruction for *Figure 2a*.

*Figure 2 continued on next page*

*Figure 2 continued*

**Source data 2.** Human PCNA 3D reconstructed map for *Figure 2b*.

**Source data 3.** Fourier shell correlation data for *Figure 2c*.

**Source data 4.** Distance histogram data and model fit for *Figure 2d*.

**Figure supplement 1.** From polarization trajectories to 2D images.

**Figure supplement 2.** Length of the segment and localization precision.

**Figure supplement 3.** Complete data set of the PCNA protein trimer used for 3D reconstruction.

**Figure supplement 3—source data 1.** Full dataset of normalized center of mass coordinates of the 2D images used for 3D reconstruction.

uncertainty as well as the random particle orientation (*Böning et al., 2021*, *Weisenburger et al., 2017*). We determine a side length of 9.9±0.6 nm in excellent agreement with the expected value. The uncertainty was determined via bootstrapping and is consistent with the resolution obtained from the FSC curve. We note that the high signal-to-noise ratio (SNR) of the method (see, e.g, *Figure 2a*), and the comparatively low information density per particle, deliver a good results from a total of 119 particles (see *Figure 1—source data 2* for overall statistics), which is two to three orders of magnitude lower than the number required for typical cryoEM measurements (*Cheng et al., 2015*). Indeed, the low SNR in cryo-EM requires data from a large number of particles to be first averaged to establish 2D classes before using them for 3D reconstruction (*Rosenthal and Henderson, 2003*). In our case, each 2D projection directly contributes to the 3D reconstruction process.

## Resolving a hexameric protein complex using partial labeling

The trimer structure discussed above involves only a single dye-dye distance. We now turn to resolving an example of more complex higher-order protein structures. Considering that a limited number of fluorophores can be resolved via stochastic blinking (*Figure 1—figure supplement 6*), we pursued a strategy of partial labeling of the sites of interest on a given individual protein complex. The piece-wise information, which involves various fluorophore arrangements and distances is then assembled to solve for the full architecture using prior knowledge of the symmetry. This concept has been successfully used to build structural models in NMR (*Fiaux et al., 2002*) and more recently in SR microscopy (*Heydarian et al., 2018*; *Molle et al., 2018*). To demonstrate this technique, we examine the homo-hexamer Caseinolytic Peptidase B (ClpB) of *Thermus thermophilus* in its quaternary structure (*Lee et al., 2003*; *Diemand and Lupas, 2006*; *Figure 3—figure supplement 1a*). ClpB is a molecular machine that rescues proteins from aggregation within cells (*Doyle et al., 2013*), and its structure was shown to be very stable in the presence of ATP at low concentrations (*Mazal et al., 2019*). Here, we labeled the M-domain of the protein at residue S428C such that the distance between two adjacent labeling sites is expected to be 9 nm after accounting for the dye linker (*Figure 3—figure supplement 1a*). We deliberately reduced the labeling efficiency to 50% in order to allow for 1/3 of the particles to carry three fluorophores as estimated by the binomial distribution (*Figure 3—figure supplement 1b*).

Labeled ClpB complexes were imaged in the same fashion as before in the presence of 2 mM ATP to stabilize the protein assembly. We obtained about 510 photons per frame on average and a significantly higher total number of photons after 100,000 frames since 87% of the molecules survived until the end of the acquisition interval (see *Figure 3—figure supplement 1c-g* for photo-physics charectrization). The photo-blinking behavior was similar to that observed in PCNA with a slight increase in the average off-on ratio, allowing robust fluorophore assignment. Hence, we fitted our polarization traces as described previously and selected only those cases that contained three fluorophores. Particles carrying three labels inherently fall into three

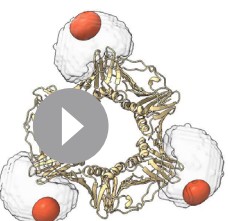

**Animation 1.** Human PCNA 3D reconstruction.

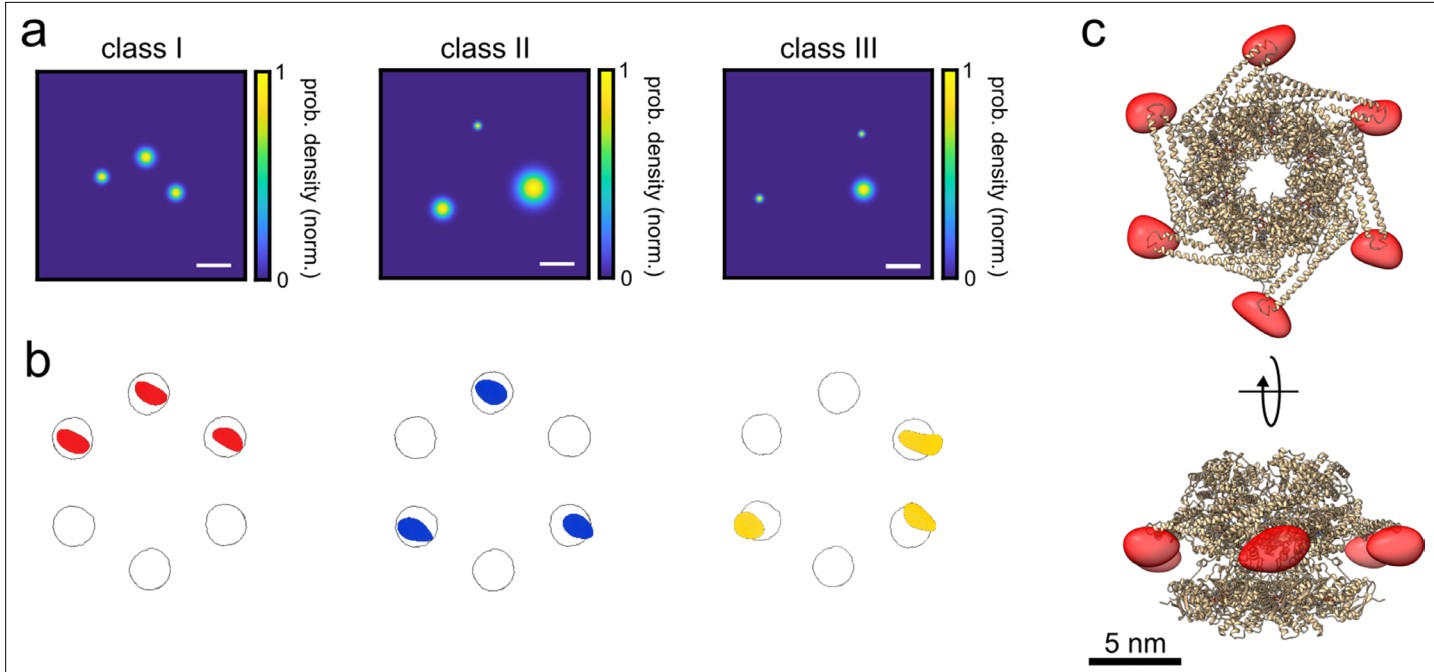

**Figure 3.** 3D reconstruction of the ClpB hexamer protein. (**a**) Top view of super-resolved 2D images for classes I, II, and III, as obtained from single-particle classification procedure. Scale bar is 3 nm. (**b**) Result of single-particle classification and averaging. The reconstructed 3D volume of each class nicely sits in the simulated accessible volume of the fluorophores (grey spheres). Red, blue and yellow spheres represent classes I, II, and III with correlation values of 0.98, 0.96, and 0.86, respectively. (**c**) 3D reconstruction of the complete hexamer obtained from merging the 3D volumes (red spheroid) of the three classes. Top figure shows the top view of the reconstructed 3D volume of the hexamer shape, and the bottom figure shows its 90° rotation (see online *Animation 2 and 3*). Crystal structure of ClpB is shown as a cartoon in gold (PDB: 1QVR) (*Lee et al., 2003*; *Diemand and Lupas, 2006*). Reconstructed 3D volumes were calculated from 232, 100, and 135 particles for classes 1, 2, and 3, respectively.

The online version of this article includes the following source data and figure supplement(s) for figure 3:

**Source data 1.** Full dataset coordinates of the 2D images used for 3D reconstruction of each class for *Figure 3a*.

**Source data 2.** Full raw dataset coordinates of the unclassified particles for *Figure 3a*.

**Source data 3.** 3D reconstituted maps of each class for *Figure 3b*.

**Source data 4.** 3D reconstituted maps of the hexamer complex for *Figure 3c*.

**Figure supplement 1.** ClpB labeling and photo-physics.

**Figure supplement 2.** Data set of the ClpB hexamer protein used for classification.

**Figure supplement 2—source data 1.** Full dataset of normalized center of mass coordinates of the 2D images used for 3D reconstruction.

**Figure supplement 3.** Estimation of particle misclassification based on simulation.

**Figure supplement 4.** Validation of the particle classification of the hexamer protein ClpB.

**Figure supplement 4—source data 1.** 3D reconstituted map of unclassified particles.

**Figure supplement 5.** FSC of the 3D volumes obtained for each class.

**Figure supplement 5—source data 1.** Fourier shell correlation data.

distinct classes (I, II, and III) with multiple pair-wise distances of approximately 9, 15, and 18 nm (*Figure 3—figure supplement 1a*).

To analyze the resulting triangle images, we used a supervised classification scheme (*Gao et al., 2004*) to assign each 2D image to one of the three classes, taking the angle of its plane into account. Here, we simulated large data sets of 2D projections for each class and applied a template matching procedure based on the 2D cross-correlation between an experimental image and the simulated images. A control based on simulated ground truth images obtained from the crystal structure of ClpB showed an accuracy of 98% in template matching. *Figure 3a* displays examples of the 2D super-resolved images of each class (see *Figure 3—figure supplement 2* for more examples of different projections, and *Figure 3—source data 1* and *Figure 3—source data 2* for full dataset). We assigned

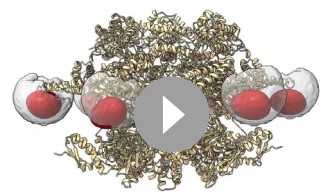

**Animation 2.** *Thermus thermophilus* ClpB 3D reconstruction (side view).

each image to the class that yielded the highest correlation score. Images that matched all classes with a difference below 10% for class 1 and 3, and below 3% for class 2 in the score (estimated from a simulation analysis, see *Figure 3—figure supplement 3*) were excluded from further analysis in order to avoid smearing (see *Figure 1—source data 2* for all statistics). Next, we picked 2D images of each class with a localization precision better than 3 nm (*Figure 3—figure supplement 1c*) and correlation score better than 0.9 and calculated their respective 3D structures as described previously for PCNA. Overall, we obtained 232, 100, and 135 particles for classes 1, 2, and 3, respectively, representing a yield of ~7% for all the particles detected with three polariza-

tion states (see *Figure 1—source data 2*). Remarkably, as illustrated in *Figure 3b*, the reconstructed 3D volumes fit very well to the probable locations of the dye on the protein structure given by the accessible volumes (circles) on each protomer after taking the dye linker into account. The correlation scores between the reconstructed 3D volumes and the accessible volumes of the dyes were 0.98, 0.96, and 0.86 for classes I, II, and III, respectively. As a control, we also performed a reconstruction of unclassified 2D images, and confirmed that no structure was identified (*Figure 3—figure supplement 4*). Again, the FSC curves of the volumes (see *Figure 3—figure supplement 5a-c*) suggest an exquisite resolution of 4.0, 7.9, and 6.4 Å, for classes I, II, and III, respectively. Following the successful assignment of the three classes, we merged them as shown in *Figure 3c* to obtain the complete 3D shape of the hexamer structure (see *Animation 2 and 3*).

An efficient classification benefits from some prior knowledge of the structure. To examine the applicability of our method for samples with unknown symmetry or side lengths, we followed a similar procedure as demonstrated by *Curd et al., 2021*. First, by inspecting the raw distance histogram of our unclassified particles obtained from single distance measurements (two polarization states), we could identify a peak at ~9 nm as the most probable side length of our molecules (see *Figure 4a*). Next, we assumed three models with different symmetries for pentamers, hexamers and heptamers, but all sharing the same side length of 9 nm. We simulated an equivalent number of projections for each model and correlated them with our experimental data. Then, we used the Akaike information criterion (*Portet, 2020*; *Curd et al., 2021*) to find the best model that describes our experimental data (see Materials and methods section). As shown in *Figure 4b*, we found that the hexamer structure matches our data significantly better than the other models. In addition, we examined a different case of hexamer model with reduced symmetry and found that this model resulted in a considerably worse fit to our data (*Figure 4b*).

Quantification of the robustness of our approach for samples without symmetry and order goes beyond the scope of our current study, but one simple strategy would be to consider a symmetric structure such as a hexamer and allow for each corner to deviate within a circle of radius R (see *Figure 4c*). Simulations show that classification of our current data becomes less robust for *R*>1 nm. We point out that solving a completely disordered structure without any prior knowledge would only be possible for complete labeling.

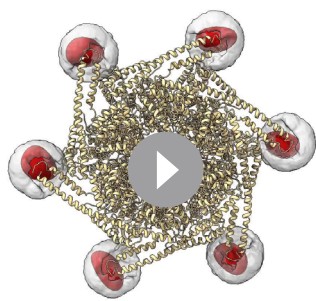

**Animation 3.** *Thermus thermophilus* ClpB 3D reconstruction (top view).

Classification and reconstruction have recently been applied to other single-molecule localization microscopy studies, and in some cases specific algorithms have been developed to handle some degree of partial labeling (*Heydarian et al., 2018*; *Sieben et al., 2018*; *Salas et al., 2017*). As shown in *Figure 4—figure supplement 1*, however, the

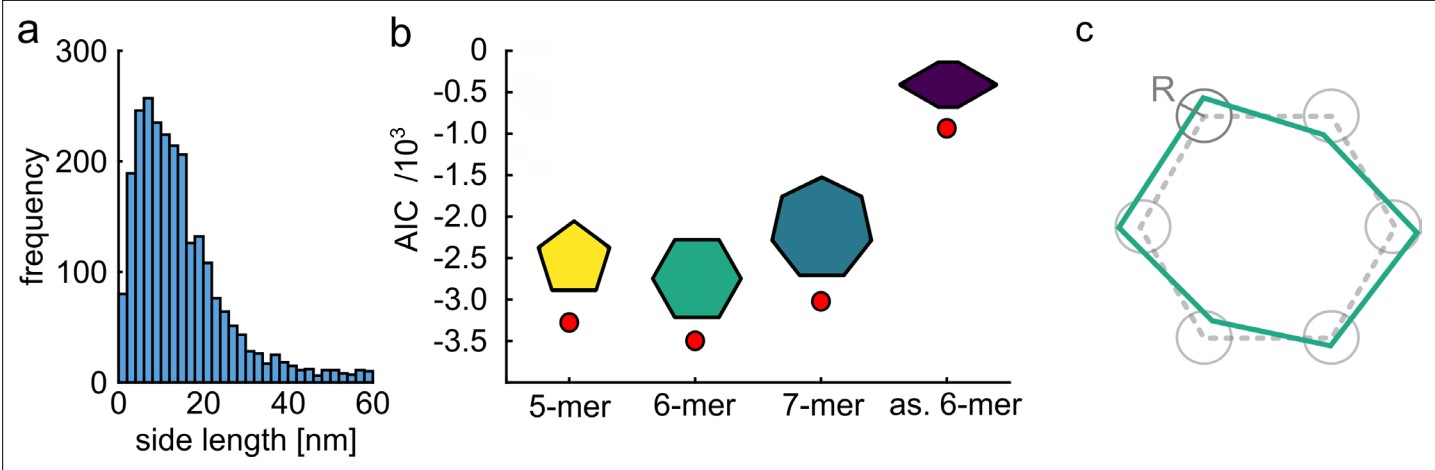

**Figure 4.** Quantitative model selection for classification. (**a**) Histogram of pair-wise distances from particles with two polarization states, showing a clear peak at ~9 nm as the most probable side length of the ClpB molecules. (**b**) Based on the identified most probable side length we built models of the oligomer with different symmetries, but sharing the same side length of 9 nm, and performed single-particle classification of the experimental images. The AIC criterion shows that the hexamer is the best model. Reducing the symmetry of the hexamer results in a worse fit. (**c**) Schematic of a model deviating from perfect symmetry. Each corner is allowed to be shifted within a circle of radius R. The classification procedure remains accurate for $R \leq$ 1 nm.

The online version of this article includes the following source data and figure supplement(s) for figure 4:

**Source data 1.** Distance histogram of the hexamer complex for *Figure 4a*.

**Figure supplement 1.** Testing different algorithms for template-free particle reconstruction.

**Figure supplement 2.** Correlative polarCOLD and cryoEM on the single particle level.

existing algorithms do not provide a satisfactory solution for our data. We attribute this to the fact that the structures in our investigations are randomly oriented in 3D, are strongly under-labeled, and involve classes that are similar. A mathematical analysis of the performance of various algorithms for structures of different morphologies is beyond the scope of our current work.

## Discussion

polarCOLD can be further improved and extended to other modalities, for example via combination with genetic labeling (*Hoffman et al., 2020*; *Dahlberg et al., 2018*; *Liu et al., 2015*; *Creemers et al., 2000*) or by exploiting the narrow absorption and emission spectra at low temperatures for co-localization of larger numbers of fluorescent markers. However, the currently achieved SNR and Angstrom resolution of cryogenic light microscopy is already capable of providing pivotal solutions for quantitative structural analysis of small proteins as well as large protein complexes and aggregates. A specially promising line of study concerns the conformation of membrane proteins in their native environment. Indeed, an estimated 20% of the human genome encodes membrane proteins and many of them are potential drug targets (*Piccoli et al., 2013*). The structure and active configuration of many proteins and protein complexes, however, remain out of reach because it is very challenging to crystallize or resolve them in their crowded surrounding (*Cheng, 2018*). In particular, heterogeneities caused by mobile domains and intrinsic distributions in assemblies cannot be preserved and classified in a robust fashion so that they lead to a smearing effect in the final reconstruction in EM studies (*Nwanochie and Uversky, 2019*; *Serna, 2019*; *Scheres, 2016*; *Scheres et al., 2007*, *Orlova and Saibil, 2004*; *Papai et al., 2020*). A future direction is, thus, to vitrify biological samples via rapid freezing (*Dubochet and McDowall, 1981*) or high-pressure freezing (*Studer et al., 2008*; *Hoffman et al., 2020*) in order to exploit the high spatial resolution, sensitivity and specificity of polarCOLD for identifying and determining the positions and orientations of individual particles of interest in low-contrast EM micrographs.

The combination of cryogenic light microscopy with cryoEM has attracted attention for cellular and tissue imaging, although the existing reports do not surpass a resolution of about 10 nm (*Dahlberg*

*and Moerner, 2021*; *Hoffman et al., 2020*; *Dahlberg et al., 2020*; *Wu et al., 2020*; *Tuijtel et al., 2019*; *Wolff et al., 2016*; *de Boer et al., 2015*; *Loussert Fonta and Humbel, 2015*; *Kaufmann et al., 2014b*, *Chang et al., 2014*; *Kaufmann et al., 2014a*, *Faas et al., 2013*; *Watanabe et al., 2011*; *Mironov and Beznoussenko, 2009*; *Agronskaia et al., 2008*; *Sartori et al., 2007*). A very exciting prospect is now to combine polarCOLD and cryoEM on the single-particle level. As illustrated in *Figure 4—figure supplement 2*, by labeling a functional domain within a single molecule with two or three fluorophores, it should be possible to classify the particles based on conformational changes as indicated by their intermolecular distances. Similarly, by labeling two different protomers or functional domains within a protein complex, it will be possible to group the particles based on intramolecular distances that result from conformational changes between subunits. In addition, while we currently screen different symmetries to find the best model that fits to the structures under study, we plan to explore unsupervised classification schemes for identifying and classifying particles in an unknown sample. In addition, one can exploit the axial information directly and follow a similar approach by *Heydarian et al., 2021*. Altogether, these approaches will help to overcome various challenges connected to low purification yields and large backgrounds from cell membranes, paving the way to studying samples containing a heterogeneous distribution of similar structures and mapping their energy landscapes.

# Materials and methods

## Key resources table

| Reagent type (species) or resource | Designation | Source or reference | Identifiers | Additional information |
|---|---|---|---|---|
| Peptide, recombinant protein | PCNA | Sigma Aldrich | Cat. #: SRP5117 | |
| Peptide, recombinant protein | ClpB | https://doi.org/10.1038/s41467-019-09474-6 | | |
| Chemical compound, drug | ATTO647N Ni-NTA | Sigma Aldrich | Cat. #: 02175–250 UG-F | |
| Chemical compound, drug | ATTO647N Maleimide | Sigma Aldrich | Cat. #: 05316–1 MG-F | |
| Chemical compound, drug | Poly-vinyl alcohol (PVA) | Sigma Aldrich | Cat. #: 360,627 | |
| Chemical compound, drug | 6-Hydroxy-2,5,7,8-tetramethylchroman-2-carboxylic Acid | TCI Deutschland GmbH | Cat. #: H0726 | |
| Software, algorithm | MATLAB 2020a | MathWorks | | |
| Software, algorithm | UCSF ChimeraX | http://www.rbvi.ucsf.edu/chimerax | | |
| Software, algorithm | DISC | DOI: 10.7554/eLife.53357 | | |
| Software, algorithm | 3D reconstruction | https://doi.org/10.1016/j.jsb.2015.03.009 | | |
| Software, algorithm | FSC | https://www.ebi.ac.uk/emdb/validation/fsc/ | | |

## Protein labeling

His-tagged human PCNA protein was purchased from Sigma Aldrich (catalogue number SRP5117), at a concentration of 6.6 μM. The protein was specifically labeled via the histidine linker on the N-terminal side of the protein with the dye ATTO647N containing a Ni-NTA functional group which was also purchased from Sigma Aldrich (catalogue number 02175–250 UG-F). First, the protein was desalted to a labeling buffer (25 mM HEPES, 25 mM KCl, pH 7.8) using a 7 K MWCO Zeba desalting column (ThermoFischer, cat. 89882) and then reacted with dyes at a ratio of 1:4 for 2 hr at RT. The protein was then desalted from the excess of dyes using the same desalting column. The labeling efficiency was estimated using an absorption spectrometer (Nanodrop 2000, ThermoFischer) confirming ~100% labeling efficiency. SDS-page and native gel indicated that the protein is indeed assembled and

labeled. N-terminal His-tagged ClpB protein, mutated at residue 428 from alanine to cysteine, was purified as described in a previous publication (*Mazal et al., 2019*). The labeling of the single cysteine was done following the same procedure as describe above with ATTO647N maleimide as a specific labeling agent of the cysteine. Absorption spectroscopy, SDS-page and native gel indicate complete assembly of the protein, see previous publication (*Mazal et al., 2019*).

## Sample preparation

Both protein samples were prepared in a similar way, and all the samples were prepared freshly on the same day. Proteins were diluted to a stock solution of 50 nM, in 25 mM HEPES, 25 mM KCl, 10 mM $MgCl_2$ 0.5 mM TCEP at pH 7.8 (working buffer). In the case of ClpB 2 mM ATP was added to stabilize protein assembly. The stock solution of poly-vinyl alcohol (PVA) was prepared as follows: 15 μl of 8% PVA (0.3% final concentration) was diluted into 345 μl working buffer containing 1 mM Trolox and 2 mM ATP in the case of ClpB. Then 2 μl of the protein was mixed with the 360 μl PVA solution and filtered with a 100 nm spin filter (Whatman Anotop-10). 5 μl of this mixed solution was then spin coated (30 s at 1000 rpm followed by 3000 rpm for 60 s) onto a plasma-cleaned mirror-enhanced substrate, prepared in house (*Böning et al., 2021*), and immediately loaded into our custom-built cryogenic microscope.

## Experimental setup

All experiments were performed in a cryogenic microscope that is built around a Janis ST-500 flow cryostat and operates at liquid helium temperature. Samples are loaded onto a cold finger and imaged by a 0.95 NA objective (Olympus MPLAPO 100 x), which is mounted in vacuum, onto two separate EMCCD cameras (Andor iXon) in a polarization-resolved configuration. The field of view spans 211 × 313 pixels with a pixel size of 190 nm. A more detailed description of the optical setup can be found in *Böning et al., 2021*. The laser intensity used in all experiments was set to ~1 $kW/cm^2$ and images were recorded with 14 ms exposure time, more than five times faster than the typical off-time. Therefore, we are able to capture individual bursts and minimize the probability of overlapping emission from two or more fluorophores in a single frame. For each field of view, we collected a total of 50,000 or 100,000 frames for PCNA and ClpB, respectively.

## Image analysis

We analyzed raw image stacks from two polarization channels with custom-written MATLAB software. Briefly, we perform dual-channel localization using maximum-likelihood estimation with a Gaussian PSF model. The minor asymmetry due to the 3D dipole orientation causes a small bias of less than 3 nm on average after filtering the data. Mechanical drift during the long acquisition times is first corrected based on image cross-correlation. Localized coordinates from both channels are then registered via an affine transformation and a non-linear correction. Registered PSFs are grouped and their polarization is calculated from the intensities. Single fluorophores are identified via this feature and used as fiducial markers to perform additional drift correction and establish a more precise image alignment via an interpolated map. The residual drift is well below 1 nm and the median registration error is less than 2 nm. Polarization time-traces of multi-fluorophore particles are then analyzed in more detail (see *Figure 1—figure supplement 3*) by a routine that is based on the DISC algorithm (*White et al., 2020*) to find the polarization states of multi-fluorophore conjugated particles. Once all polarization states are identified, the associated coordinates are averaged by taking their median to generate a 2D super-resolved image. We took particles which were fitted best to three polarization states and used them for further analysis (see *Figure 1—source data 2* for statistics).

## Single-molecule 2D image fitting

To fit the 2D maps of the human PCNA, we used the theoretical distances obtained from the structural model of human PCNA (PDB: 1AXC) and then we performed a least-squares fit over rotation angles and translation using a simulated annealing algorithm. Two-dimensional cross-correlation of the experimental and simulated maps yield a score better than 0.92. In the case of the hexamer protein, ClpB, we generated a large space of multiple 2D projections, ~25,000 images for each class, using a rotation matrix over x, y, and z axis with an angular resolution of 10°. The exact distances were evaluated from the real crystal structural model taking the dye linker into account (ClpB model, PDB:

1QVR, and the full assembly model taken from *Diemand and Lupas, 2006*). Then we performed 2D template matching between experimental data and all the images from the simulated data. Here, we used a 2D cross-correlation algorithm to calculate the similarity index between two images. The correlation score ranges from 0 to 1, where 1 indicates a 100% match. For each experimental image, we select the fit with the highest score as the best fit to our data, and to assign it to a class. Images that were fitted to all classes with a difference below 10% for classes 1 and 3, and 3% for class 2 in the score were excluded from further analysis to avoid smearing (see *Figure 3—figure supplement 3*).

## Tomography and 3D reconstruction

We selected the particles which were fitted properly to the simulated 2D projections, that is particles with a cross-correlation score better than 0.9 and a localization precision better than 3 nm. The 2D maps of the particles were then normalized (maximum and width) to perform the 3D reconstruction. For a full 3D reconstruction, we used the subspaceEM algorithm (*Dvornek et al., 2015*) with default settings and 100 runs. The algorithm was initialized with an elliptical Gaussian placed in the center of the volume. The 2D maps of the trimer protein, PCNA, were constructed using a grid size of 120 × 120 with a pixel size of 1.5 Å. In the case of the hexamer protein, the grid size was 200 × 200 with a pixel size of 1.5 Å. The 3D volumes were then processed, fitted to the crystal structures/maps of the dye location and edited using ChimeraX (*Goddard et al., 2018*). The Fourier shell correlation curves (FSC), were computed using the freely available FSC server, provided by the Protein Data Bank in Europe website (https://www.ebi.ac.uk/pdbe/emdb/validation/fsc/). Here we divide the 2D images into two equally large data sets, and calculate their 3D reconstruction as described above. Then we align the two reconstructed volumes and calculate the FSC and determine the resolution based on half-bit criteria (*van Heel and Schatz, 2005*).

## Quantitative model selection using the Akaike Information Criterion (AIC)

To show that our method can infer the proper symmetry of a protein complex, we followed a similar approach as demonstrated by *Curd et al., 2021*, but taking the random projections into account. First, we simulated multiple projections of a regular pentamer, hexamer and heptamer, all with the same side length of 9 nm. Given their symmetry, we obtain 2, 3, and 4 classes when considering three randomly placed fluorophores, respectively. We also simulated a deformed hexamer (as. 6-mer) composed of two isosceles triangles of 9 nm length on opposing sides of a square with a side length of 4 nm. Here, we need to consider a total of 5 classes. Then, we performed 2D cross correlation of the experimental data with these templates, and we picked the particles with a localization precision better than 3 nm and correlation score better than 0.5. The normalized sum of square residuals (nSSR) for each model is calculated as

$$nSSR = \sum_{j=1}^{N} \frac{\sum_{i=1}^{n}(1-x_i)^2}{n}$$

where j is the number of classes, n is the number of images per class, and $x_i$ is the score of the classified image. Then, we assessed how well each model fits the data by calculating its AIC value (*Portet, 2020*; *Curd et al., 2021*) via $AIC = M log(nSSR) + 2K$, where M is the total number of particles in each model and K is the number of parameters (8, 10, 12, 13 for the pentamer, hexamer, heptamer and deformed hexamer, respectively). K was calculated based on the number of sides in each model, number of distances, and number of classes. For example, the pentamer has 5 sides, 1 distance, and 2 classes.

## Distance fitting

For more quantitative analysis of the pair-wise distances we used a model described in *Weisenburger et al., 2017*; *Böning et al., 2021*. In short, we convolve the Rician distribution of the distance between two spots with finite localization uncertainty with the projection onto the image plane given by a cosine function. In the case of the PCNA trimer, we only need to consider a single distance due to the symmetry of the structure.

## Determination of expected measurement yield depending on angular resolution

Our co-localization procedure relies on identifying the unique polarization signature belonging to one of a maximum of N fluorophores within a diffraction limited area. To this end, we can choose to consider the complete distribution of polarization values and identify well-separated peaks (histogram method), or consider polarization information in the time domain and identify recurring levels and transitions between them (DISC algorithm *White et al., 2020*). Both methods are limited in resolution, that is depending on the signal-to-noise ratio and switching rates there is a smallest difference between any two fluorophore dipole orientations that we are able to resolve. We consider a particle with N fluorophores to be completely resolved if all N-1 pairwise angular separations are above this angular resolution. Thus, the yield of completely resolved particles depends on how well two polarization states can be distinguished. We performed a Monte Carlo simulation to estimate this yield in our measurements. To this end, we generate large sets of N random fluorophore angles in the interval [0°,90°] and compute the pairwise differences. For a given angular resolution s between 0.1° and 90° we then calculate the fraction of particles for which all differences are larger than s. This simple estimation shows that for a case of just 2 fluorophores an angular resolution of 20° is already sufficient to achieve a yield of 50%. The same yield for the case of 6 fluorophores, however, requires an angular resolution of 2°.

## Acknowledgements

We thank Tobias Utikal for assistance with cryogenic experiments, reading and commenting on the manuscript and Michelle Küppers for reading and commenting on the manuscript. We also thank Houman Mirzaalian Dastjerdi and Harald Koestler for fruitful discussions. We thank Simone Ihloff for support with materials, Alexander Gumann for preparing the mirror-enhanced substrates and Jan Renger for help with measuring the PVA film thickness. We are further grateful to Prof. Gilad Haran for the generous gift of ClpB proteins and plasmids. HM was supported by the Planning & Budgeting Committee of the Council of Higher Education of Israel. We are grateful to the Max Planck Society for financial support.

## Additional information

### Funding

| Funder | Grant reference number | Author |
|---|---|---|
| Planning and Budgeting Committee of the Council for Higher Education of Israel | 2020 - VATAT Postdoctoral fellowships for outstanding candidates from the Arab sector | Hisham Mazal |
| Max Planck Society | | Hisham Mazal Franz-Ferdinand Wieser Vahid Sandoghdar |

The funders had no role in study design, data collection and interpretation, or the decision to submit the work for publication.

### Author contributions

Hisham Mazal, Conceptualization, Data curation, Designed the experiment, Designed the experiment and carried out the measurements, Formal analysis, Investigation, Methodology, Software, Validation, Visualization, Writing - original draft, Writing - review and editing; Franz-Ferdinand Wieser, Data curation, Designed and prepared the optical setup, Formal analysis, Investigation, Methodology, Software, Validation, Visualization, Writing - original draft, Writing - review and editing; Vahid Sandoghdar, Conceptualization, Designed the experiment, Funding acquisition, Investigation, Methodology, Project administration, Supervision, Validation, Visualization, Writing - review and editing

## Author ORCIDs
Hisham Mazal http://orcid.org/0000-0002-2071-9552
Vahid Sandoghdar http://orcid.org/0000-0003-2594-4801

## Decision letter and Author response
Decision letter https://doi.org/10.7554/eLife.76308.sa1
Author response https://doi.org/10.7554/eLife.76308.sa2

## Additional files

### Supplementary files
• Transparent reporting form

### Data availability
Raw data generated or analyses during this study are included in the manuscript file and supporting files. Source data files have been provided for Figure 1b, c, Figure 1 - figure supplement 1 and 3. Figure 2a, b, c, d, Figure 2 - figure supplement 3. Figure 3a, b, c, Figure 3 - figure supplement 2,4, 5. Figure 4a.

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
