## [Editor Report]

This paper will be of interest to the structural biology community and people working on cryogenic fluorescence microscopy. This paper is a clear step forward in the use of single-molecule localization microscopy at Å resolution, thanks to low-temperature polarized super-resolution imaging and advanced data processing algorithms.

---

## [Decision Letter]

**Decision letter after peer review:**

Thank you for submitting your article "Deciphering a hexameric protein complex with Å optical resolution" for consideration by *eLife*. Your article has been reviewed by 3 peer reviewers, and the evaluation has been overseen by a Reviewing Editor and Volker Dötsch as the Senior Editor. The following individual involved in review of your submission has agreed to reveal their identity: Sophie Brasselet (Reviewer #1).

Essential revisions:

All three reviewers of this manuscript have pointed out important questions that I'm convinced will help the authors to resubmit a revised paper that is much stronger and clearer. The detailed reviews are enclosed in this email, and I would strongly encourage the authors to address all the questions and comments raised in those reports. That being said, the most important points are listed here (see referee reports for the details):

1) About the non-Gaussian shape of a 3D-oriented dipole PSF: what is its effect on localization precision?

2) Discuss the limitations of the method: requirement for no dipole rotation, orientation discrimination, etc.

3) Discuss a more general/wider applicability of the method: how can this be applied to non-symmetric/unknown complexes. How necessary is it the use of particle symmetry to determine the structures?

4) Discuss the reliability of particle discrimination based on the fluorophore polarization state.

5) Provide a better/fairer comparison of this technique to EM (e.g. number of needed particles, low yield in determination, etc.)

6) Expand on the methods (especially because this is a methods paper), in particular in the image analysis bit.

*Reviewer #1 (Recommendations for the authors):*

Some points, detailed here, need to be addressed to clarify the findings, their conclusion and the claims.

1. Effect of the 3D orientation of the fluorophores.

1.1 Even though the used NA is low and the incident polarization is in the sample plane, the localization precision required here is extreme and necessarily comes in competition with the inacuracy of PSF-centroid determination when the dipole is oriented in 3D. In particular, PSFs from 3D dipoles are not symetric Gaussians anymore. It is quite surprising that the 3D-orientation bias does not surpass, in some situations, the other bias sources in the reported data. This bias needs to be estimated and introduced in the localization accuracy estimation, together with the other inaccuracy sources (noise, drift).

1.2 Is the presence of the mirror-enhanced substrate modifying the expected brightness from dipoles at different axial positions, as well as the 3D-orientation bias mentioned above?

2. Orientational and time emission dynamics of the fluorophores.

2.1 The orientation of the dipoles is considered fixed at low T. The authors should give the expected number for the rotational diffusion of the fluorophore. We would expect that if some degree of rotational motion is permitted by the linker between the dipole and protein, this could be an issue for the reconstruction of the information, even at low T since more orientations will be probable for the measured dipoles.

2.2 It is not clear how the time dynamics of emission of the fluorophores affects the reliability of the discrimination of the different oriented dipoles in the time traces recorded. Is there any need for fine-tuning this on-off dynamics as well as the recording time of the measurement such as only one dipole emits at a time ? Optimal settings rules should be mentioned.

3. Orientational discrimination of fluorophores.

3.1 The authors mention in line 126 “The maximum number of resolvable polarization states per protein is currently limited to fewer than 5, dictated by the width of the polarization histogram”

How is this number 5 extracted from the width of the histograms, and how does this number depend on the signal level, possible presence of orientational mobility (see above)?

3.2 Would this number increase if the measurement also includes the quantification of the 3D orientation of the dipoles as mentioned below? If the measurement also includes spectral discrimination in addition to polarization discrimination?

3.3 The previous work by the authors (Boning et al.,) involved important signal thresholding, which was determining in the final dipole-distances assessment. How necessary is the signal thresholding here and what how is the choice of this filtering performed?

3.4 The authors mention in line 128 the necessity to exclude cases where the expected number of polarization states is not well resolved. What is the origin of this effect and is this also a limit of the unsupervised statistical learning algorithm used to treat the time traces?

3.5 The authors mention an angular resolution required for resolving 3 vs 6 fluorophores in a structure. How is the experimental angular resolution measured and what are the experimental sources of errors that can affect this resolution?

4. Classification and reconstruction.

4.1 The fluorophores assignment relies on a robust supervised classification procedure and template matching, which decreases the number of unknowns. How would this classification perform if the structure would contain unknown parameters?

4.2 In Figure 3 —figure supplement 2, it is not visible by eye that some of these figures belong to one class rather than another: some of them look very similar (e.g. aligned 3 dots) and belong to different classes.

4.3 Table S1 shows that the yield to reconstruct 3D information is very low (for ClpB, 100-200 particles are used out of more than 23000 particles detected initially). A lot of particles are also excluded from the analysis because a low-confidence score (line 202). What would be needed to increase this yield and increase the confidence score?

5. Methods

5.1 Is the fact that there is an ambiguity on the projection angle retrieval a problem? Since the orientation measured is confined in the [0-90{degree sign}] range, can't the orientation discrimination be problematic if a whole angle sector is missed in the angles determination?

5.2 How is the 100 photon/frame threshold (Figure S3 legend) chosen, what is the rationale behind in terms of estimation precision and accuracy? Isn't intensity thresholding also excluding 3D dipoles orientations?

5.3 The authors mention in line 160 a model to extract the side lengths of the projected triangles, what are the detail and hypotheses of this model?

5.4 The 3D protein orientation is linked to the spatial repartition of the fluorophores, however there is no mention of the possibility to use the 3D z position of the fluorophores to infer such a 3D information. Can't this axial position be exploited?

5.5 There are several recent techniques developed to infer the 3D orientation from emitting dipoles (using PSF engineering and/or polarization splitting, possibly using radial-polarization filtering). The measurement of the 3D orientation of the fluorophores would be of great value to complete the information with an additional angular parameter, which could increase the number of measurable fluorophores. The authors should comment on this, and possibly test this extension of the approach.

5.6 The abstract (line 43) states that the method “promises to provide crucial insight into intrinsic, environmental and dynamic heterogeneities of biomolecular structures.” The present method requires a large amount of data to be collected in the context of a known, fixed protein structure which density is highly controlled. It seems therefore very robust in the limited framework of low-level labelling, immoblilized and low concentration proteins, however it is difficult to envision its application in a native, dynamic configuration where diffusion potentially comes into play. The authors should provide more detail about what would be the ingredients needed to enlarge the application range.

*Reviewer #2 (Recommendations for the authors):*

In Weissenburger 2017, the authors already presented the reconstruction of four fluorophore sites within streptavidin. Overall, I think the manuscript presents an advance of previous developments by the authors, but the generalizability and applicability to more complex samples remain partly open. This could be discussed in more detail.

1) Fluorophore discrimination depends on the correct assignment of the different polarization states. It would be good to include a representation of the uncertainties in fluorophore assignment and the resulting filtering procedure. For clarity, it would also help to color-code the different fluorophores as identified from the polarization state in the scatter plots. For example in Figure 2A, some spots appear larger or have an irregular shape, which could be associated with the uncertainty (as mentioned above) or the result of overlapping molecules for which a color code could help.

2) I am wondering about how generalizable the described reconstruction approach is or can be with respect to non-symmetric complexes or mixed samples of multiple protein species. To what extent does the reconstruction rely on prior knowledge of the investigated proteins? For example, the simulations used to classify the triangle images in the case of ClpB. What are the boundaries of these simulations (distance, angle, etc)? Is it possible to combine this with existing pattern extraction schemes (i.e. Curd, 2020)? Also, would more classes be helpful in increasing the precision or applicability of the approach? Particle reconstruction typically relies on many more classes unless some of them are redundant due to symmetries.

*Reviewer #3 (Recommendations for the authors):*

The paper by Mazal, Wieser and Sandoghdar presents a method to image molecular complexes with light microscopy, but still obtain resolutions/localization uncertainties of fluorescent labels that have been limited to cryo EM.

This submission build upon earlier work from the same group (Weisenburger 2017 and Boning 2021). The polarisation detection does not increase the sparsity compared to Weisenburger, but on the detection side they can identify different emitters based on their fixed dipole emission as shown in Boning. They further improve the sparsity, by explicit under labelling which later is compensated by particle registration and averaging. Compared to Boning the submission adds 3D reconstructions of two molecular complexes from 2D under labelled structures. The polarisation and localization method is improved technically, but the concept was already there.

The presented methodology and results are overall very nice.

In the following a few remarks and details that could help to improve the manuscript or that are unclear to me.

– Abstract: many it would good to directly state to how many sites you extend the method (l34).

In how far was the use of a supervised classification and use of particle symmetry really needed? (l93) Typically in cryoEM the use of symmetry is only needed to increase the resolution or in the absence of a large number of particles. What does the method deliver if the symmetry is not implied? That is a strong prior knowledge assumption and hinders discovery of new insights. The supervised classification is similar as model assumptions could be brought into the reconstruction.

In the introduction it would be nice to mention the idea of Hafi et al., Fluorescence nanoscopy by polarization modulation and polarization angle narrowing. Nature Methods, 11:579-584, 2014. and also Hulleman et al., Fluorescence polarization control for on-off switching of single molecules at cryogenic temperatures. Small Methods, page 1700323, 2018. Similar ideas to exploit polarisation of the fixed dipole emitters have been introduced here too.

l112 the polarisation state give the in-plane dipole angle. Would the method benefit from an estimation of the polar angle in addition? around l125 this could increase the number of resolvable polarisation states, but this might not be the limiting factor but the sparsity of the blinking?

l132 After localizations and identifying emitters by their polarisation state, how is the average position computed? Is that done weighted by the photon count?

l162 The remark the only 119 particles are needed for a reconstruction and not many more as in cryo EM is a bit misplaced. The information density in the LM is very low (only the few coordinates of the particles), while the EM map is a full image with millions of pixels that contain information. In addition early LM particle averaging results have used also hundredth of particles only typically. In addition the benefit of increasing the number of particles in LM is diminishing, because the additional particles cannot improve the localization precision but only fill in missing information thereby increasing the SNR. For cryo EM the resolution scale as sqrt(N) with number of particles, for LM this is much worse and eventually zero.

l181 Could you provide the t_on and t_off for the emitters. The 50% underlabelling is based on their ratio?

l203 "better 3 nm" -> better than 3 nm.

l205 yield of 7,46% point and comma not correct. 2 digits behind comma not needed. In addition why was the yield so low? Even if you consider that the polarisation state removes 3/4 of the particles the yield is still low. In cryoEM most particles needs to be removed due to damage resulting from stress close to the air water interface. What could be the reason here?

l 217 The claim to "all possible orientations" is a bit strong. I do not see why the 3 citations, in particular Huijben and Sieben, cannot deal with data in all possible orientations. I would argue that up to now, either the particles in a top-down configuration have been images because the structure was 2D or in the case of the much used NPC because the nuclear envelope is running nearly parallel to the cover slip. Therefore the NPC only have a limited tilt range. In conclusion, the claim to being the first is a big too much, as this is certainly not a methodical advance.

l277 the amount of Trolox 1mM seems quite a lot. Is this needed to suppress triplet states?

l288 What is the physical pixel size of the camera. The model is not stated, but the back projected size is stated as 190 nm with a 100x objective. This information is also not present in the supplement of Boning2021.

l290-298 Image analysis.

This part is vague on the details. Is the code available? I think as the analysis is not standard this needs to be open. The Poisson-weighted Gaussian MLE, what is that? Fitting a Gaussian function with Poisson likelihood? Will not polarisation effects of the restricted dipole play a role already at 0.95NA? (see Stallinga 2010, Optics Express 18:24461 and Engelhardt 2010, NanoLetters 11:209). The drift correction is non-linear, but for the rest it is totally unclear how it works. However, that is quite important as the acquisition time is a bit less than 1h.

[Editors’ note: further revisions were suggested prior to acceptance, as described below.]

Thank you for resubmitting your article "Deciphering a hexameric protein complex with Å optical resolution" for consideration by *eLife*. Your article has been reviewed by 2 peer reviewers, and the evaluation has been overseen by a Reviewing Editor and Volker Dötsch as the Senior Editor. The reviewers have opted to remain anonymous.

Essential revisions:

1) Please, consider answering the 3 specific comments from Reviewer #3, which need no new experiments, just clarification in the text and maybe some simple extra data analysis.

*Reviewer #2 (Recommendations for the authors):*

The authors have done well in responding to my comments, which at least partly came from misreading and/or misunderstanding on my side. The paper remains a difficult read but shows a clear step forward in using low temperature to increase SNR in single-molecule imaging.

*Reviewer #3 (Recommendations for the authors):*

Overall the authors did a very nice job with the revision. I would support the publication of the manuscript.

point (2) I strongly disagree that classification is needed and an assumption of symmetry. Many published averaging techniques for LM deal exactly with the problem described by the authors "In that sense, the incompletely labelled object inherently behaves as a heterogenous sample with multiple labelling configurations and distances". This is why most of the cryo-EM algorithms fail when applied to strongly under labelled data, they see the different labelling states as different classes. Please have a look at the work of Heydarian et al., (Nat Methods 2018, Nat Com 2021), Sieben et al., (Nat Methods 2018), Salas et al., (PNAS 2017), Shi et al., (PLOS one 2019) and maybe others.

The question at hand is, if the under labelling here is so severe that indeed the above methods cannot deliver a good reconstruction. From my own experience somewhere between 30-50% degree of labelling is needed to obtain a meaningful reconstruction. However, this depends also on the absolute number of sites. Some of the above papers have open access code, it might be worthwhile to try this at least.

The approach to validate their model via the AIC is good. The idea from Curd et al., have been well incorporated as this method typically only can learn something for the nearest neighbour distance, exactly what it has been used for her.

point (5) The average best position of a set of localisations from the same molecule is typically done by photon weighted averaging not median computation? If there are strong outliers which suggest the use of the median, maybe it would be better to remove the outliers first and then compute a weighted average?

point (13) I see that the author fit a 2D Gaussian, at the expense of a bit of blurring due to model mismatch. I would expect that the random dipole orientation do not average about but give rise to a large standard deviation of the localization than strictly needed with a correct model fitting. The localization will be unbiased due to the randomness, but it will result in a bit of extra blurring.

---

## [Author Response]

Essential revisions:All three reviewers of this manuscript have pointed out important questions that I’m convinced will help the authors to resubmit a revised paper that is much stronger and clearer. The detailed reviews are enclosed in this email, and I would strongly encourage the authors to address all the questions and comments raised in those reports. That being said, the most important points are listed here (see referee reports for the details):1) About the non-Gaussian shape of a 3D-oriented dipole PSF: what is its effect on localization precision?2) Discuss the limitations of the method: requirement for no dipole rotation, orientation discrimination, etc.

We address these points in section 1, 2 and 3 of Reviewer 1.

3) Discuss a more general/wider applicability of the method: how can this be applied to non-symmetric/unknown complexes. How necessary is it the use of particle symmetry to determine the structures?

We address this point in section 4 of Reviewer 1, section 1 and 3 of Reviewer 2, and section 2 of Reviewer 3.

4) Discuss the reliability of particle discrimination based on the fluorophore polarization state.

We discuss this point in sections 2, 3 and 5 of Reviewer 1, point 2 of Reviewer 2, and point 4 of Reviewer 3.

5) Provide a better/fairer comparison of this technique to EM (e.g. number of needed particles, low yield in determination, etc.)

We discuss this point in section 4 of Reviewer 1, section 3 of Reviewer 2, and sections 2, 6 and 9 of Reviewer 3.

6) Expand on the methods (especially because this is a methods paper), in particular in the image analysis bit.

We address this issue in section 13 of Reviewer 3

Reviewer #1 (Recommendations for the authors):Some points, detailed here, need to be addressed to clarify the findings, their conclusion and the claims.1. Effect of the 3D orientation of the fluorophores.1.1 Even though the used NA is low and the incident polarization is in the sample plane, the localization precision required here is extreme and necessarily comes in competition with the ymmetric of PSF-centroid determination when the dipole is oriented in 3D. In particular, PSFs from 3D dipoles are not ymmetric Gaussians anymore. It is quite surprising that the 3D-orientation bias does not surpass, in some situations, the other bias sources in the reported data. This bias needs to be estimated and introduced in the localization accuracy estimation, together with the other inaccuracy sources (noise, drift).

The reviewer points to an important consideration. The localization bias due to the asymmetry of the dipole PSF is a major contribution to the overall localization uncertainty. However, the overall extent of this contribution is reduced by the fact that we preferentially excite the in-plane components and detect them more efficiently due to the lower NA. In addition, by discriminating against less bright fluorescence signals, we eliminate contributions from molecules that have dipole moments with substantial axial component. We estimate the contribution to the localization inaccuracy to be no more than 3 nm on average as was discussed in our recent publication (Böning et al., ACS Photonics 2021). Nevertheless, it is possible to reach subnanometer resolution with a more selective choice of the particles that are included in the averaging process, as validated in Figure 2b-d and Figure 3b.

In the revised manuscript, we extended the image analysis part of the methods section to include our estimates for the contributions to the localization uncertainty.

1.2 Is the presence of the mirror-enhanced substrate modifying the expected brightness from dipoles at different axial positions, as well as the 3D-orientation bias mentioned above?

Yes, the effect of the mirror is two-fold. It increases the power radiated into the upper half space of the objective depending on the axial position and, thus, affects the collection efficiency. Moreover, due to interference of the incoming plane wave with the reflected wave the intensity of illumination is modulated in the axial direction. This creates an optical sectioning effect such that the brightness of fluorophores also depends on their axial position with respect to this standing wave pattern. We choose the spacer thickness on top of the mirror to be approximately a quarter-wavelength, such that the first antinode coincides with the spacer-sample interface. As mentioned in the main text (line 95) we refer the readers to our recent publication (Böning et al., ACS Photonics 2021) for more details on the properties of the mirror substrate.

L95: “see (Böning et al., 2021 for in-depth characterization of the mirror-enhanced substrate).

2. Orientational and time emission dynamics of the fluorophores.2.1 The orientation of the dipoles is considered fixed at low T. The authors should give the expected number for the rotational diffusion of the fluorophore. We would expect that if some degree of rotational motion is permitted by the linker between the dipole and protein, this could be an issue for the reconstruction of the information, even at low T since more orientations will be probable for the measured dipoles.

We assume the orientation of dipoles to be fixed since the environment of the fluorophore is in a fully frozen solid state. Polarization-resolved imaging supports this assumption since for single fluorophores we find shot-noise limited polarization angles that remain constant over time. The shot noise limit of our polarization signal is ca. 4-5 degrees (see Author response image 1). Moreover, for single dye molecules we do not observe any jumps or diffusion of their polarization angle during 25 min of measurement time. This is fully in line with the wisdom of more than three decades of low temperature single-molecule spectroscopy. Thus, we can safely conclude that the rotational diffusion is less than a few degrees based on the angular resolution. This information is mentioned in the following passage of the main text, including the figure as well.

L118: “The number of fluorophores that can be simultaneously resolved depends on the blinking on-off dynamics (see method section). Furthermore, the shot noise determines the angular resolution and, thus, the maximum number of resolvable polarization states per protein. This, in turn, directly affects the yield for resolved particles (see Figure 1 —figure supplement 5-6). For example, in our current experiment, we used one polarization basis and projected all orientations to the limited space of θ ϵ [0°, 90°]. By taking the experimental angular resolution of ca. 5° (Figure 1 —figure supplement 5-6), we can theoretically expect to resolve 50% of the particles which contain 4 fluorophores, and roughly 15% of the particles which contain 6 fluorophores.”

**Author response image 1. sa2fig1:** Polarization histogram of single fluorophore ATTO 647N imaged at 4K. The histogram shows single population with a width of 5 degrees, which matches the expected shotnoise value estimated from 100 photons.

2.2 It is not clear how the time dynamics of emission of the fluorophores affects the reliability of the discrimination of the different oriented dipoles in the time traces recorded. Is there any need for fine-tuning this on-off dynamics as well as the recording time of the measurement such as only one dipole emits at a time? Optimal settings rules should be mentioned.

The dynamics of emission dictates the admissible fluorophore density and the necessary imaging speed. As a rule of thumb, to resolve N labels unambiguously within a diffraction-limited spot, we require an on-off ratio of less than 1/N. In addition, the frame rate of the camera needs to be faster than the average off-time to minimize the probability of overlapping contributions from many molecules in a single image. Unfortunately, it is generally not straightforward to predict the on- and off-times of fluorophores at cryogenic temperatures from their values in solution and common strategies cannot be directly used to engineer them. While we have confirmed an intensity dependence of the on-times in isolated fluorophores, the overall statistics still depends critically on the local environment. Here, in accordance with previous work at cryogenic temperatures (Weisenburger et al., Nat. Methods 2017; Hoffman et al., Science 2020), we find exceptionally long dark states. Since N is less than a handful of labels in our case, both criteria stated above are fulfilled. For the protein-fluorophore conjugates investigated in this work we found average on- and off-times of 17 ms and 73 ms, respectively. Therefore, an exposure time of 14 ms is sufficient to resolve single fluorophore blinking. We now mention this in the experimental setup part of the methods section.

L320: “The laser intensity used in all experiments was set to ~1 kW/cm2 and images were recorded with 14 ms exposure time, more than 5 times faster than the typical off-time. Therefore, we are able to capture individual bursts and minimize the probability of overlapping emission from two or more fluorophores in a single frame.”

3. Orientational discrimination of fluorophores.3.1 The authors mention in line 126 “The maximum number of resolvable polarization states per protein is currently limited to fewer than 5, dictated by the width of the polarization histogram”How is this number 5 extracted from the width of the histograms, and how does this number depend on the signal level, possible presence of orientational mobility (see above)?

Since fluorophores are orientated randomly with respect to each other, their polarization angles can only be distinguished if they are separated by more than the angular resolution given by the associated shot-noise, i.e., the angular resolution depends on the average number of photons per localization. Since fluorophores show a variation in their brightness, the angular resolution in assessing their dipoles also varies. In figure supplement 2 of Figure 1, we show the obtained total photons per frame. Taking an average of 100 photons, we arrive at an upper bound for the width of ca. 3-4 degrees. As stated before, our measurements of single fluorophores show that we reach this shot-noise limit (Author response image 1), suggesting no influence of orientational mobility. In figure supplement 6 of Figure 1 we performed a simulation to determine the fraction of particles for which all fluorophore polarizations are separated by more than the angular resolution. At a resolution of 5 degrees, we would obtain less than 30% yield beyond 5 fluorophores before filtering. This is indeed not a strict theoretical limitation. We have now removed this strict statement and only explain the important conditions for 3D localization.

3.2 Would this number increase if the measurement also includes the quantification of the 3D orientation of the dipoles as mentioned below? If the measurement also includes spectral discrimination in addition to polarization discrimination?

The reviewer makes very interesting suggestions, which are on the to-do list of future experiments. Including the 3D orientation of dipoles and/or their spectral information would certainly increase the dimensions along which fluorophores can be distinguished and would therefore scale up our method. Indeed, we could capitalize on the advantage of fixed orientation as well as narrower linewidths at low temperatures. However, acquiring more information comes at the expense of more sophisticated experimental protocols and analysis as well as longer integration times.

3.3 The previous work by the authors (Boning et al.,) involved important signal thresholding, which was determining in the final dipole-distances assessment. How necessary is the signal thresholding here and what how is the choice of this filtering performed?

The distribution of single dye-dye distances assumes an asymmetric shape when the localization uncertainty becomes comparable to the distance itself. In our previous work we demonstrated the narrowing of this distribution for a dye-dye separation of 6.5 nm by filtering the data for the brightest emitters. Here, the filtering process is less strict since the smallest distance we resolve is around 9 nm. The exact choice of the filtering process, however, also depends on the desired or required localization precision. In our current work, we could resolve the distance histogram (Figure 2 d) with a precision of 0.6 nm by selecting the particles with a localization precision below 3 nm. This particle selection comes over the yield, while in (Boning et al., 2020) the yield was ~ 60% , here we obtain ~ 7 %.

3.4 The authors mention in line 128 the necessity to exclude cases where the expected number of polarization states is not well resolved. What is the origin of this effect and is this also a limit of the unsupervised statistical learning algorithm used to treat the time traces?

There is no single algorithm which can achieve 100 % accuracy on the estimation of the number of polarization states. Therefore, in some cases the algorithm might over or under-estimate the number of polarization states, thus affecting the number of detected molecules. In particular, if the projections of two polarization are very close, or if the SNR is very low it might not be possible to distinguish them in a reliable fashion. In our case, we prefer to use an unsupervised algorithm as it is very user friendly and adapted very well to our pipeline. We modified our statement in the main text accordingly:

L118:” The number of fluorophores that can be simultaneously resolved depends on the blinking on-off dynamics (see method section). Furthermore, the shot noise determines the angular resolution and, thus, the maximum number of resolvable polarization states per protein. This, in turn, directly affects the yield for resolved particles (see Figure 1 —figure supplement 5-6). For example, in our current experiment, we used one polarization basis and projected all orientations to the limited space of θ ϵ [0°, 90°]. By taking the experimental angular resolution of ca. 5° (Figure 1 —figure supplement 5-6), we can theoretically expect to resolve 50% of the particles which contain 4 fluorophores, and roughly 15% of the particles which contain 6 fluorophores. Adding a second polarization basis at a tilt of 45°, would allow one to double the angular space (Stallinga and Rieger, 2012). Moreover, using polarized illumination (Backer et al., 2016, Zhanghao et al., 2019) and a full 3D characterization of the dipole moments (Lieb et al., 2004, Mortensen et al., 2010, Hulleman et al., 2021) would result in less overlap and thus enhanced capacity for identifying different polarization states. We remark, however, that even in our current scheme, one can improve the number of resolved polarization states by excluding the ambivalent cases, albeit at the expense of the overall yield (see Figure 1 —figure supplement 6 and Table supplement 1 for the statistics of this analysis).”

3.5 The authors mention an angular resolution required for resolving 3 vs 6 fluorophores in a structure. How is the experimental angular resolution measured and what are the experimental sources of errors that can affect this resolution?

Unlike the brightness, which is subject to fluctuations due to time-dependent changes in the blinking behaviour, the lifetime or the quantum yield, the polarization angle is remains unchanged. We estimate the experimental angular resolution based on the number of photons per frame (see figure supplement 2 of Figure 1 and Author response image 1) to be ca. 5 degrees. However, we want to emphasize that this is an average value and some fluorophores will show considerably broader polarization features. In particular, those with a polarization close to either 0° or 90° are not visible in one channel, narrowing the actual range of detection. It should also be noted that even 6 or more fluorophores could be resolved with our method if one is willing to accept the lower fraction of such particles extracted from a measurement.

4. Classification and reconstruction.4.1 The fluorophores assignment relies on a robust supervised classification procedure and template matching, which decreases the number of unknowns. How would this classification perform if the structure would contain unknown parameters?

We would like to clarify that the fluorophore assignment does not depend on any model it is done using DISC algorithm in a model-free manner. However, the arrangement of the fluorophores on the protein is determined using prior knowledge/initial models. In general, if we have a homogenous sample, such as the trimer protein PCNA which adopts a single conformation only, or a homogenously and fully labelled object (Weisenburger et al., Nat. Methods 2017), our method would work without any prior assumptions. It should, of course, be evident that more data might be necessary to collect.

However, the challenge we want to address in this work is particle classification of an incompletely labelled system and reconstruction of a high-order protein complex based on that. In that sense, the incompletely labelled object inherently behaves as a heterogenous sample with multiple labelling configurations and distances, as described for the hexamer protein (class 1, class 2 and class 3). In addition, multiple different 3D orientations of proteins make the problem even more challenging. Thus, to classify these particles and converge to the true structure one needs to assume an initial model. To the best of our knowledge, currently, there are no algorithms that solve such an issue in a model-free manner (unsupervised). Even in the cryo-EM community it is common practice to work with existing structures as initial models for classification and alignment. Thus, the fact that we can nevertheless distinguish and resolve the three classes at high resolution, considering the subtle changes between them, is still remarkable. In fact, this establishes an important future step for our method and toward combination with other structural methods such as cryo-EM.

To address the reviewer concern, we reduced the number of known parameters, following a similar approach to (Curd et al., 2020), as also mentioned by Reviewer 2 and 3. Here we show that the symmetry assumptions could indeed be relaxed. First, by inspecting the raw distance histogram of our unclassified particles (Author response image 2) we could identify a peak of ~ 9 nm as the major side length of our protein complex. Second, we assumed three different models, with different symmetries (pentamer, hexamer, and heptamer), but sharing the same side length of 9 nm (author response image 2b). Then we cross-correlated our experimental images with these different models and calculated the best model which fitted our data using the Akaike information criteria (AIC), as discussed now in the method section. Interestingly, we found that the best model is a symmetric hexamer (Author response image 2). In addition, we examined a different case of hexamers. We simulated a deformed shape of hexamer structure, which composed of two side lengths 9 and 4 nm (Author response image 2), and we found that this model fitted our experimental data significantly worse (Author response image 2). Next, we also tried to test how minor deviations from a perfect hexamer could affect the classification procedure. Here, we found that changes up to ± 1 nm could still successfully classify our experimental data (Author response image 2). Given these results, we believe that our approach does not rely on a strict assumption/knowledge of the particle symmetry since one can blindly screen different initial models for classification, and find the best choice which describes the experimental data. Quantification of the robustness conditions for a general particle with limited prior knowledge is an interesting mathematical task, which is beyond the scope of our current work.

To address the reviewer’s concerns, we have modified the main text as follows:

L225: “Efficient classification benefits from some prior knowledge of the structure. To examine the applicability of our method for samples with unknown symmetry or side lengths, we followed a similar procedure as demonstrated by (Curd et al., 2021). First, by inspecting the raw distance histogram of our unclassified particles obtained from single distance measurements (two polarization states), we could identify a peak at ~9 nm as the most probable side length of our molecules (see Figure 4a). Next, we assumed three models with different symmetries for pentamers, hexamers and heptamers, but all sharing the same side length of 9 nm. We simulated an equivalent number of projections for each model and correlated them with our experimental data. Then, we used the Akaike information criterion (Portet, 2020, Curd et al., 2021) to find the best model that describes our experimental data (see method section). As shown in Figure 4b, we found that the hexamer structure matches our data significantly better than the other models. In addition, we examined a different case of hexamer model with reduced symmetry and found that this model resulted in a considerably worse fit to our data (Figure 4b).

Quantification of the robustness of our approach for samples without symmetry and order goes beyond the scope of our current study, but one simple strategy would be to consider a symmetric structure such as a hexamer and allow for each corner to deviate within a circle of radius R (see Figure 4c). Simulations show that classification of our current data becomes less robust for R>1 nm. We point out that solving a completely disordered structure without any prior knowledge would only be possible for complete labelling.”

**Author response image 2. sa2fig2:** Quantitative model selection for classification. (a) Histogram of pair-wise distances from particles with two polarization states, showing a clear peak at ~9 nm as the most probable side length of the ClpB molecules. (b) Based on the identified most probable side length we built models of the oligomer with different symmetries, but sharing the same side length of 9 nm, and performed single-particle classification of the experimental images. The AIC criterion shows that the hexamer is the best model. Reducing the symmetry of the hexamer results in a worse fit. (c) Schematic of a model deviating from perfect symmetry. Each corner is allowed to be shifted within a circle of radius R. The classification procedure remains accurate for R<=1 nm.

4.2 In Figure 3 —figure supplement 2, it is not visible by eye that some of these figures belong to one class rather than another : some of them look very similar (e.g. aligned 3 dots) and belong to different classes.

The reviewer is right in that all the classes presented here are similar to some extent, for example all of them share at least one similar side length. For example, Class 1 and Class 2 share similar distances of 15 nm. Class 1 and 3 share the distances 9 and 15 nm and so on. If our projection were top views only, this would be obvious for us and we would not need any type of supervised classification. However, due to multiple orientations of the particles, the side lengths are somewhat distorted (looks shorter in some cases) and thus, it hard to judge by eye which particle is related to which class. In addition, all classes contain triangles, so that at certain projections they would look similar.

As verified by simulations, the 2D cross correlation algorithm turns out to be powerful in determining the class of each particle. The classification based on simulation yield 98% accuracy. Among the 2% of the misclassified particles, results showed that 46% were wrong assignment of particles from class 3 into class 1, 23% of particles form class 3 into 2, 23 % of class 2 into 1, and 8 % are of class 1 into 3. By examining the correlation score of these misclassified particles, we found that they had very similar scores. As seen in Figure 3 —figure supplement 3, some have a negligible score of 0 and others are up to 5 %. However, as described in the main text, we include information from particles with a very high correlation score above 0.9, and at the same time, we exclude particles with score difference below 10 %. Thus, theoretically, we should be able to exclude them completely from our classification and 3D reconstruction. We would like to note that the 2% misclassification is due to a number of simulated projections. In fact, we found that by increasing the number of projections (finer steps of rotational degrees) we could reduce misclassifications to 0.5%, leading a difference between the scores that is essentially 0.

We now clarify the sentence regarding to this and add a supplement figure:

L207: “Images that were fitted to all classes with a difference below 10% for class 1 and 3, and below 3% for class 2 (estimated from a simulation analysis, see Figure 3 —figure supplement 3) in the score were excluded from further analysis in order to avoid smearing.”

4.3 Table S1 shows that the yield to reconstruct 3D information is very low (for ClpB, 100-200 particles are used out of more than 23000 particles detected initially). A lot of particles are also excluded from the analysis because a low-confidence score (line 202). What would be needed to increase this yield and increase the confidence score?

The harsh reduction from ~ 23000 particles into ~ 6000, in the case of ClpB, has different origins. First, as described in the main text, we started by 50% labelling of the protein complexes. As shown in Figure 3 —figure supplement 1 b, 70% of the particle are disqualified as they are labeled with 1, 2, or 4-6 fluorophores. Second, some particles are also missed during the time trace fitting using DISC algorithm, for example, particles assigned to two polarization states rather than three. Third, proteins might be damaged during preparation and handling (e.g., spin coating). Fourth, although the photophysics is generally improved at low temperatures, some fluorophores bleach spontaneously. In addition, as pointed out in point 5.1, overlap of polarization projections can also contribute to reducing our yield.

The reduction of particles from 6000 particles to ~500 particles happens as a result of filtering: (1) Based on the correlation score (here we consider particles with a correlation score above 0.9). (2) Based on localization precision (here we consider particles with a localization precision better than 3 nm). (3) We further filter particles with a correlation score difference of up to 10% for class 1 and 3, and 3% for class 2. In fact, it is hard to judge the true yield, as here we rely on the performance of the DISC algorithm, which dictates the number of particles with three polarizations (as mentioned in point 3.4). Furthermore, we exclude particles where the SNR of the trace fit falls below 2. In the trimer case we might also be confronted with another issue. The labelling in that case was done on the histidine tag linker, which is not covalently bound, and might thus disassemble once we dilute the protein.

To increase the yield, one can take several measures. First, one could improve the assignment of fluorophore identification with better algorithms, or by including more information such as spectral information or resolving the 3D orientation of each fluorophore. For example, in the case of ClpB, specific labelling of each monomer in the complex would allow one to achieve full labelling with three fluorophores, thus increasing our yield. However, this approach is still very challenging and out of reach. In fact, we can also turn into fully labeling of the complex and exploit the 4-5 labeled molecules as well in order to gain more information. Second, regarding the classification, at the moment we select the best particles which match our model structure. In practice, we can increase our library for template matching, to include 1 nm variation at each distance, which would increase our yield. Lastly, one could expect to improve the sample quality by switching to 100% aqueous medium, such as vitreous ice, as used in the cryo-EM community.

5. Methods5.1 Is the fact that there is an ambiguity on the projection angle retrieval a problem? Since the orientation measured is confined in the [0-90{degree sign}] range, can't the orientation discrimination be problematic if a whole angle sector is missed in the angles determination?

The reviewer is right to point out a limitation of the polarization-resolved detection. As we only measure in one polarization basis, all angles are mapped into the first quadrant. The resulting ambiguity reduces the range over which fluorophores can be distinguished by their dipole orientation. In principle, one can do better by measuring in two different polarization bases to extend the range to [0°,180⁰], and by extracting the 3D orientation of each fluorophore, e.g. by engineering the PSF. However, these approaches generally require a higher numerical aperture and signal. We now point these out in the manuscript.

L118:” The number of fluorophores that can be simultaneously resolved depends on the blinking on-off dynamics (see method section). Furthermore, the shot noise determines the angular resolution and, thus, the maximum number of resolvable polarization states per protein. This, in turn, directly affects the yield for resolved particles (see Figure 1 —figure supplement 5-6). For example, in our current experiment, we used one polarization basis and projected all orientations to the limited space of θ ϵ [0°, 90°]. By taking the experimental angular resolution of ca. 5° (Figure 1 —figure supplement 5-6), we can theoretically expect to resolve 50% of the particles which contain 4 fluorophores, and roughly 15% of the particles which contain 6 fluorophores. Adding a second polarization basis at a tilt of 45°, would allow one to double the angular space (Stallinga and Rieger, 2012). Moreover, using polarized illumination (Backer et al., 2016, Zhanghao et al., 2019) and a full 3D characterization of the dipole moments (Lieb et al., 2004, Mortensen et al., 2010, Hulleman et al., 2021) would result in less overlap and thus enhanced capacity for identifying different polarization states. We remark, however, that even in our current scheme, one can improve the number of resolved polarization states by excluding the ambivalent cases, albeit at the expense of the overall yield (see Figure 1 —figure supplement 6 and Table supplement 1 for the statistics of this analysis).”

5.2 How is the 100 photon/frame threshold (Figure S3 legend) chosen, what is the rationale behind in terms of estimation precision and accuracy? Isn't intensity thresholding also excluding 3D dipoles orientations?

The threshold for photons/frame is a trade-off between localization precision and accuracy of photon assignment. While a higher value excludes frames from analysis, possibly lowering the localization precision, it also increases the accuracy of assignment as the remaining frames have lower shot noise and can therefore be assigned with higher confidence. Dipoles with larger outof-plane angles are more likely to be completely excluded, which is, however, desirable given their larger localization bias. The exact threshold of 100 photon/frame (total counts from both cameras) was manually set as a reasonable compromise. We now add a figure in the supplement which shows the relation between localization precision as a function of the segment length (number of frames for each fluorophore). It shows that the longer the segment, the higher the localization precision.

5.3 The authors mention in line 160 a model to extract the side lengths of the projected triangles, what are the detail and hypotheses of this model?

The fit shown in Figure 2d is based on the following model: The distance distribution between two points in a plane, each determined with a finite localization uncertainty, is mathematically described by a so-called Rician function. If the expected distance value is large compared to the localization uncertainty it resembles a regular Gaussian. In our measurement, however, particles are randomly oriented within the sample, such that the distance between two localized fluorophores is a projection of the real distance. We include this projection by convolving the Rician with a cosine function, which results in the characteristic shape shown in Figure 2d. In the case of the PCNA trimer, we only need to consider a single distance due to the symmetry of the structure. We add this sentence in the relevant method section.

5.4 The 3D protein orientation is linked to the spatial repartition of the fluorophores, however there is no mention of the possibility to use the 3D z position of the fluorophores to infer such a 3D information. Can't this axial position be exploited?

We currently infer the third dimension by averaging many projections. This comes with its own complications as pointed out by all reviewers. By directly localizing fluorophores in 3D we could circumvent this step and facilitate our analysis. Instead of a 3D reconstruction, we could then perform particle alignment directly on the localization. However, most options for 3D localization (e.g. astigmatism, double helix, Vortex etc.) will enlarge the PSF and therefore reduce the SNR. Furthermore, the axial position is usually determined with lower precision than the lateral coordinates. As more groups join the emerging efforts in cryogenic super-resolution microscopy, we expect these variations to be examined in different arrangements.

5.5 There are several recent techniques developed to infer the 3D orientation from emitting dipoles (using PSF engineering and/or polarization splitting, possibly using radial-polarization filtering). The measurement of the 3D orientation of the fluorophores would be of great value to complete the information with an additional angular parameter, which could increase the number of measurable fluorophores. The authors should comment on this, and possibly test this extension of the approach.

We thank the reviewer for bringing up this issue. Measuring full 3D dipole orientation indeed adds information that improves the applicability of our method to more complex situations, while also eliminating the known dipole bias issue in a rigorous way. However, these approaches generally reduce the SNR because the emitted photons are either split between more channels or more pixels. Radial polarization as recently demonstrated by the group of W.E. Moerner decrease the signal by 30% even for perfectly horizontal dipoles. Nevertheless, we consider full 3D orientation estimation to be a valuable addition to be explored further. We have added the following statement to the manuscript to reflect this:

L125: “Adding a second polarization basis at a tilt of 45°, would allow one to double the angular space (Stallinga and Rieger, 2012). Moreover, using polarized illumination (Backer et al., 2016, Zhanghao et al., 2019) and a full 3D characterization of the dipole moments (Lieb et al., 2004, Mortensen et al., 2010, Hulleman et al., 2021) would result in less overlap and thus enhanced capacity for identifying different polarization states. We remark, however, that even in our current scheme, one can improve the number of resolved polarization states by excluding the ambivalent cases, albeit at the expense of the overall yield (see Figure 1 —figure supplement 6 and Table supplement 1 for the statistics of this analysis).”

5.6 The abstract (line 43) states that the method “promises to provide crucial insight into intrinsic, environmental and dynamic heterogeneities of biomolecular structures.” The present method requires a large amount of data to be collected in the context of a known, fixed protein structure which density is highly controlled. It seems therefore very robust in the limited framework of low-level labelling, immoblilized and low concentration proteins, however it is difficult to envision its application in a native, dynamic configuration where diffusion potentially comes into play. The authors should provide more detail about what would be the ingredients needed to enlarge the application range.

First, we would like to clarify that our usage of the word dynamic was unfortunate in that we did not mean to refer to temporally changing events. Rather, we wanted to emphasize conformational heterogeneity that might have happened due to dynamic processes prior to our sample preparation. We have now modified the abstract to avoid misunderstandings.

Our method is complementary to single particle cryo-EM methodology in the sense that both methods work on homogenous and diluted samples of interest. While cryo-EM measurements suffer from the sensitivity of electron interactions with biological matter, in our case we enjoy a high sensitivity that is intrinsic to the large cross section of light-matter interaction. Thus, here, we can use the information of known structures to resolve heterogenous ligand binding pockets or localizing protein ligand bindings within a protein complex, a realm which is challenging for cryo-EM. Second, as we show in the revised manuscript (Figure 4), our method can also tackle protein complexes with lower degree of symmetry, and less a priori knowledge of the structure to be resolved. Third, we believe that our approach would indeed be capable of resolving membrane proteins in their native environment, such as cell membrane, which is challenging for other biophysical methods, such as X-ray and cryo-EM. By preserving the sample in vitreous ice, we would be able to combine our optical classification scheme with cryo-EM studies, thus combining the strengths of the two methods. As demonstrated in our sample, optical particle classification can be fed to single particle cryo-EM to resolve particle orientation and conformation at high sensitivity. By doing so multiple conformational states can be identified and completely resolved (on the level of amino acids) at high resolution.

Reviewer #2 (Recommendations for the authors):In Weissenburger 2017, the authors already presented the reconstruction of four fluorophore sites within streptavidin. Overall, I think the manuscript presents an advance of previous developments by the authors, but the generalizability and applicability to more complex samples remain partly open. This could be discussed in more detail.

A similar point was also raised by Reviewer 1. We attach our answer to points 4.1 and 5.6 of Reviewer 1 below and we address the question in depth in point 3:

In general, if we have a homogenous sample, such as the trimer protein PCNA which adopts a single conformation only, or a homogenously and fully labelled object (Weisenburger et al., Nat. Methods 2017), our method would work without any prior assumptions. It should, of course, be evident that more data might be necessary to collect.

However, the challenge we want to address in this work is particle classification of an incompletely labelled system and reconstruction of a high-order protein complex based on that. In that sense, the incompletely labelled object inherently behaves as a heterogenous sample with multiple labelling configurations and distances, as described for the hexamer protein (class 1, class 2 and class 3). In addition, multiple different 3D orientations of proteins make the problem even more challenging. Thus, to classify these particles and converge to the true structure one needs to assume an initial model. To the best of our knowledge, currently, there are no algorithms that solve such an issue in a model-free manner (unsupervised). Even in the cryo-EM community it is common practice to work with existing structures as initial models for classification and alignment. Thus, the fact that we can nevertheless distinguish and resolve the three classes at high resolution, considering the subtle changes between them, is still remarkable. In fact, this establishes an important future step for our method and toward combination with other structural methods such as cryo-EM.

Our method is complementary to single particle cryo-EM methodology in the sense that both methods work on homogenous and diluted samples of interest. While cryo-EM measurements suffer from the sensitivity of electron interactions with biological matter, in our case we enjoy a high sensitivity that is intrinsic to the large cross section of light-matter interaction. Thus, here, we can use the information of known structures to resolve heterogenous ligand binding pockets or localizing protein ligand bindings within a protein complex, a realm which is challenging for cryo-EM. Second, as we show in the revised manuscript (Figure 4), our method can also tackle protein complexes with lower degree of symmetry, and less a priori knowledge of the structure to be resolved. Third, we believe that our approach would indeed be capable of resolving membrane proteins in their native environment, such as cell membrane, which is challenging for other biophysical methods, such as X-ray and cryo-EM. By preserving the sample in vitreous ice, we would be able to combine our optical classification scheme with cryo-EM studies, thus combining the strengths of the two methods. As demonstrated in our sample, optical particle classification can be fed to single particle cryo-EM to resolve particle orientation and conformation at high sensitivity. By doing so multiple conformational states can be identified and completely resolved (on the level of amino acids) at high resolution.

1) Fluorophore discrimination depends on the correct assignment of the different polarization states. It would be good to include a representation of the uncertainties in fluorophore assignment and the resulting filtering procedure. For clarity, it would also help to color-code the different fluorophores as identified from the polarization state in the scatter plots. For example in Figure 2A, some spots appear larger or have an irregular shape, which could be associated with the uncertainty (as mentioned above) or the result of overlapping molecules for which a color code could help.

We thank the reviewer for his/her suggestions. We now provide the residuals of the DISC fit to better represent the uncertainty associated with this step (Figure 1 main text). Also, we provide a colour code of two different trajectories, and mention the associated localization precision for each fluorophore. In addition, we attach a figure which shows the localization precision as a function of the length of each assigned polarization (number of frames). As expected, the longer the segment, the higher the achievable localization precision. We refer the readers to these two new figures as well (Figure 2 —figure supplement 1-2).

Regarding the super-resolved projection images in Figure 2a, we can unfortunately not follow the Reviewer’s concern. All but the last image in Figure 2a, as well as the majority of images in the corresponding figure supplement 1, show three clearly separated spots and we describe in the figure caption that the spot size is given by the localization precision of the respective fluorophore. In some cases, the shape looks elongated as a result of two overlapping spots.

2) I am wondering about how generalizable the described reconstruction approach is or can be with respect to non-symmetric complexes or mixed samples of multiple protein species. To what extent does the reconstruction rely on prior knowledge of the investigated proteins? For example, the simulations used to classify the triangle images in the case of ClpB. What are the boundaries of these simulations (distance, angle, etc)? Is it possible to combine this with existing pattern extraction schemes (i.e. Curd, 2020)? Also, would more classes be helpful in increasing the precision or applicability of the approach? Particle reconstruction typically relies on many more classes unless some of them are redundant due to symmetries.

We have answered similar question raised by Reviewer 1 (point 4.1) and have revised the manuscript:

To address the reviewer concern, we reduced the number of known parameters, following a similar approach to (Curd et al., 2020), as mentioned by the Reviewer. Here we show that the symmetry assumptions could indeed be relaxed. First, by inspecting the raw distance histogram of our unclassified particles (Author response image 2) we could identify a peak of ~ 9 nm as the major side length of our protein complex. Second, we assumed three different models, with different symmetries (pentamer, hexamer, and heptamer), but sharing the same side length of 9 nm (Author response image 2). Then we cross-correlated our experimental images with these different models and calculated the best model which fitted our data using the Akaike information criteria (AIC), as discussed now in the method section. Interestingly, we found that the best model is a symmetric hexamer (Author response image 2). In addition, we examined a different case of hexamers. We simulated a deformed shape of hexamer structure, which composed of two side lengths 9 and 4 nm (Author response image 2), and we found that this model fitted our experimental data significantly worse (Author response image 2). Next, we also tried to test how minor deviations from a perfect hexamer could affect the classification procedure. Here, we found that changes up to ± 1 nm could still successfully classify our experimental data (Author response image 2). Given these results, we believe that our approach does not rely on a strict assumption/knowledge of the particle symmetry since one can blindly screen different initial models for classification, and find the best choice which describes the experimental data. Quantification of the robustness conditions for a general particle with limited prior knowledge is an interesting mathematical task, which is beyond the scope of our current work.

L225: “Efficient classification benefits from some prior knowledge of the structure. To examine the applicability of our method for samples with unknown symmetry or side lengths, we followed a similar procedure as demonstrated by (Curd et al., 2021). First, by inspecting the raw distance histogram of our unclassified particles obtained from single distance measurements (two polarization states), we could identify a peak at ~9 nm as the most probable side length of our molecules (see Figure 4a). Next, we assumed three models with different symmetries for pentamers, hexamers and heptamers, but all sharing the same side length of 9 nm. We simulated an equivalent number of projections for each model and correlated them with our experimental data. Then, we used the Akaike information criterion (Portet, 2020, Curd et al., 2021) to find the best model that describes our experimental data (see method section). As shown in Figure 4b, we found that the hexamer structure matches our data significantly better than the other models. In addition, we examined a different case of hexamer model with reduced symmetry and found that this model resulted in a considerably worse fit to our data (Figure 4b).

Quantification of the robustness of our approach for samples without symmetry and order goes beyond the scope of our current study, but one simple strategy would be to consider a symmetric structure such as a hexamer and allow for each corner to deviate within a circle of radius R (see Figure 4c). Simulations show that classification of our current data becomes less robust for R>1 nm. We point out that solving a completely disordered structure without any prior knowledge would only be possible for complete labelling.”

Regarding the comment on the classes. There is some difference between the classes we define here and the definition of classes raised by the reviewer. In cryo-EM, due to a very low SNR, they perform class average of particles which share the same in-plane orientation in order to increase the SNR for 3D reconstruction. Thus, the number of classes is defined by the total number of inplane orientation/projections. In our case, we do not need any class average, as our SNR is already very high, and these particles can be fed immediately into 3D reconstruction. This avoids the introduction of wrong classifications and some noise into the system. The three classes we define here as in the case of the hexamer protein, are based on conformational state/heterogeneity within the same protein complex. Each class here (act as different sample) adopts different 3D orientation and has different distances. As we have shown, give some degree of prior knowledge, we could identify these classes/conformations and resolve them separately in 3D. Such an approach can be combined with cryo-EM to help identifying particle orientation and heterogeneity with the same protein complex to resolve each state at high resolution without smearing.

We now mention this in the text:

L169: “Indeed, the low SNR in cryo-EM requires data from a large number of particles to be first averaged to establish 2D classes before using them for 3D reconstruction (Rosenthal and Henderson, 2003). In our case, each 2D projection directly contributes to the 3D reconstruction process.”

Reviewer #3 (Recommendations for the authors):The paper by Mazal, Wieser and Sandoghdar presents a method to image molecular complexes with light microscopy, but still obtain resolutions/localization uncertainties of fluorescent labels that have been limited to cryo EM.This submission build upon earlier work from the same group (Weisenburger 2017 and Boning 2021). The polarisation detection does not increase the sparsity compared to Weisenburger, but on the detection side they can identify different emitters based on their fixed dipole emission as shown in Boning. They further improve the sparsity, by explicit under labelling which later is compensated by particle registration and averaging. Compared to Boning the submission adds 3D reconstructions of two molecular complexes from 2D under labelled structures. The polarisation and localization method is improved technically, but the concept was already there.The presented methodology and results are overall very nice.In the following a few remarks and details that could help to improve the manuscript or that are unclear to me.– Abstract: many it would good to directly state to how many sites you extend the method (l34).

We now state clearly to how many sites we extend this method.

“We showcase this method (polarCOLD) by resolving the trimer arrangement of proliferating cell nuclear antigen (PCNA) and six different sites of the hexamer protein Caseinolytic Peptidase B (ClpB) of Thermus thermophilus in its quaternary structure, both with Å resolution.”

In how far was the use of a supervised classification and use of particle symmetry really needed? (l93) Typically in cryoEM the use of symmetry is only needed to increase the resolution or in the absence of a large number of particles. What does the method deliver if the symmetry is not implied? That is a strong prior knowledge assumption and hinders discovery of new insights. The supervised classification is similar as model assumptions could be brought into the reconstruction.

Similar points were raised by Reviewer 1 (point 4.1) and Reviewer 2 (point 1 and 3). We attach our answer again for convenience:

In general, if we have a homogenous sample, such as the trimer protein PCNA which adopts a single conformation only, or a homogenously and fully labelled object (Weisenburger et al., Nat. Methods 2017), our method would work without any prior assumptions. It should, of course, be evident that collecting more data might be necessary.

However, the challenge we want to address in this work is particle classification of an incompletely labelled system and reconstruction of a high-order protein complex based on that. In that sense, the incompletely labelled object inherently behaves as a heterogenous sample with multiple labelling configurations and distances, as described for the hexamer protein (class 1, class 2 and class 3). In addition, multiple different 3D orientations of proteins make the problem even more challenging. Thus, to classify these particles and converge to the true structure one needs to assume an initial model. To the best of our knowledge, currently, there are no algorithms that solve such an issue in a model-free manner (unsupervised). Even in the cryo-EM community it is common practice to work with existing structures as initial models for classification and alignment. Thus, the fact that we can nevertheless distinguish and resolve the three classes at high resolution, considering the subtle changes between them, is still remarkable. In fact, this establishes an important future step for our method and toward combination with other structural methods such as cryo-EM.

To address the reviewer concern, we reduced the number of known parameters, following a similar approach to (Curd et al., 2020), as also mentioned by Reviewer 2. Here we show that the symmetry assumptions could indeed be relaxed. First, by inspecting the raw distance histogram of our unclassified particles (Author response image 2) we could identify a peak of ~ 9 nm as the major side length of our protein complex. Second, we assumed three different models, with different symmetries (pentamer, hexamer, and heptamer), but sharing the same side length of 9 nm (Author response image 2). Then we cross-correlated our experimental images with these different models and calculated the best model which fitted our data using the Akaike information criteria (AIC), as discussed now in the method section. Interestingly, we found that the best model is a symmetric hexamer (Author response image 2). In addition, we examined a different case of hexamers. We simulated a deformed shape of hexamer structure, which composed of two side lengths 9 and 4 nm (Author response image 2), and we found that this model fitted our experimental data significantly worse (Author response image 2). Next, we also tried to test how minor deviations from a perfect hexamer could affect the classification procedure. Here, we found that changes up to ± 1 nm could still successfully classify our experimental data (Author response image 2). Given these results, we believe that our approach does not rely on a strict assumption/knowledge of the particle symmetry since one can blindly screen different initial models for classification, and find the best choice which describes the experimental data. Quantification of the robustness conditions for a general particle with limited prior knowledge is an interesting mathematical task, which is beyond the scope of our current work.

To address the reviewer’s concerns, we have modified the main text as follows:

L225: “Efficient classification benefits from some prior knowledge of the structure. To examine the applicability of our method for samples with unknown symmetry or side lengths, we followed a similar procedure as demonstrated by (Curd et al., 2021). First, by inspecting the raw distance histogram of our unclassified particles obtained from single distance measurements (two polarization states), we could identify a peak at ~9 nm as the most probable side length of our molecules (see Figure 4a). Next, we assumed three models with different symmetries for pentamers, hexamers and heptamers, but all sharing the same side length of 9 nm. We simulated an equivalent number of projections for each model and correlated them with our experimental data. Then, we used the Akaike information criterion (Portet, 2020, Curd et al., 2021) to find the best model that describes our experimental data (see method section). As shown in Figure 4b, we found that the hexamer structure matches our data significantly better than the other models. In addition, we examined a different case of hexamer model with reduced symmetry and found that this model resulted in a considerably worse fit to our data (Figure 4b).

Quantification of the robustness of our approach for samples without symmetry and order goes beyond the scope of our current study, but one simple strategy would be to consider a symmetric structure such as a hexamer and allow for each corner to deviate within a circle of radius R (see Figure 4c). Simulations show that classification of our current data becomes less robust for R>1 nm. We point out that solving a completely disordered structure without any prior knowledge would only be possible for complete labelling.”

In the introduction it would be nice to mention the idea of Hafi et al., Fluorescence nanoscopy by polarization modulation and polarization angle narrowing. Nature Methods, 11:579-584, 2014. and also Hulleman et al., Fluorescence polarization control for on-off switching of single molecules at cryogenic temperatures. Small Methods, page 1700323, 2018. Similar ideas to exploit polarisation of the fixed dipole emitters have been introduced here too.

We thank the reviewer for these suggestions. We have inserted these citations in our revised manuscript.

L75: “Control of the fluorescence signal via polarization modulation has also been shown to offer an alternative to random blinking (Hulleman et al., 2018, Hafi et al., 2014).”

l112 the polarisation state give the in-plane dipole angle. Would the method benefit from an estimation of the polar angle in addition? around l125 this could increase the number of resolvable polarisation states, but this might not be the limiting factor but the sparsity of the blinking?

The reviewer is correct. By also measuring the polar angle we could increase the space for discriminating fluorophores based on their dipole orientation (a 3-fold improvement considering the available angles in a quarter circle vs. a half-sphere). But the reviewer is also right in recognizing that the limiting factor is indeed the blinking kinetics. To significantly scale the density of fluorophores would require much lower on-off ratios than we currently observe.

l132 After localizations and identifying emitters by their polarisation state, how is the average position computed? Is that done weighted by the photon count?

The calculation of the localized position is done with a median.

l162 The remark the only 119 particles are needed for a reconstruction and not many more as in cryo EM is a bit misplaced. The information density in the LM is very low (only the few coordinates of the particles), while the EM map is a full image with millions of pixels that contain information. In addition early LM particle averaging results have used also hundredth of particles only typically. In addition the benefit of increasing the number of particles in LM is diminishing, because the additional particles cannot improve the localization precision but only fill in missing information thereby increasing the SNR. For cryo EM the resolution scale as sqrt(N) with number of particles, for LM this is much worse and eventually zero.

The reviewer raises an interesting and subtle point. Indeed, our method delivers far less information about the structure than would be obtained from a cryoEM density map, i.e. we obtain a similar resolution but only for the locations of a few labelling sites. Perhaps unsurprisingly, we therefore require fewer particles to obtain a high-resolution reconstruction. Nevertheless, we wanted to raise awareness for this conceptual difference between LM and cryoEM. Following the reviewer’s justified remark, we attempted a better formulation of our original statement.

We now address this issue in the manuscript, where we write:

L165:“ We note that the high signal-to-noise ratio (SNR) of the method (see, e.g, Figure 2a), and the comparatively low information density per particle, deliver a good results from a total of 119 particles (see Table supplement 1 for overall statistics), which is two to three orders of magnitude lower than the number required for typical cryoEM measurements (Cheng et al., 2015). Indeed, the low SNR in cryo-EM requires data from a large number of particles to be first averaged to establish 2D classes before using them for 3D reconstruction (Rosenthal and Henderson, 2003). In our case, each 2D projection directly contributes to the 3D reconstruction process.”

l181 Could you provide the t_on and t_off for the emitters. The 50% underlabelling is based on their ratio?

We provide now the on time (17 ms) and the off time (73 ms) of the emitters. The 50% labeling was used intentionally to achieve a high percentage of 3 fluorophores/particle as described in the main text and in Figure 3 —figure supplement 1 b.

l203 "better 3 nm" -> better than 3 nm.

Thanks, we corrected this typo.

l205 yield of 7,46% point and comma not correct. 2 digits behind comma not needed. In addition why was the yield so low? Even if you consider that the polarisation state removes 3/4 of the particles the yield is still low. In cryoEM most particles needs to be removed due to damage resulting from stress close to the air water interface. What could be the reason here?

We answered this question in Reviewer 1 comment, point 4.3:

The harsh reduction from ~ 23000 particles into ~ 6000, in the case of ClpB, has different origins. First, as described in the main text, we started by 50% labelling of the protein complexes. As shown in Figure 3 —figure supplement 1 b, 70% of the particle are disqualified as they are labeled with 1, 2, or 4-6 fluorophores. Second, some particles are also missed during the time trace fitting using DISC algorithm, for example, particles assigned to two polarization states rather than three. Third, proteins might be damaged during preparation and handling (e.g., spin coating). Fourth, although the photophysics is generally improved at low temperatures, some fluorophores bleach spontaneously. In addition, as pointed out in point 5.1, overlap of polarization projections can also contribute to reducing our yield.

The reduction of particles from 6000 particles to ~500 particles happen as a result of filtering: (1) Based on the correlation score (here we consider particles with a correlation score above 0.9). (2) Based on localization precision (here we consider particles with a localization precision better than 3 nm). (3) We further filter particles with a correlation score difference of up to 10% for class 1 and 3, and 3% for class 2. In fact, it is hard to judge the true yield, as here we rely on the performance of the DISC algorithm, which dictates the number of particles with three polarizations (as mentioned in point 3.4). Furthermore, we exclude particles where the SNR of the trace fit falls below 2. In the trimer case we might also be confronted with another issue. The labelling in that case was done on the histidine tag linker, which is not covalently bound, and might thus disassemble once we dilute the protein.

We note that cryo-EM studies have similarly low yields of 10-30% (Merk et al., 2016), as a result of iterative brute-force filtration at the 2D class average steps, and consequently at the 3D classification process. This not explicitly due to the damage of molecules at the air water interface, but also from other factors as well, mainly particle heterogeneity.

l 217 The claim to "all possible orientations" is a bit strong. I do not see why the 3 citations, in particular Huijben and Sieben, cannot deal with data in all possible orientations. I would argue that up to now, either the particles in a top-down configuration have been images because the structure was 2D or in the case of the much used NPC because the nuclear envelope is running nearly parallel to the cover slip. Therefore the NPC only have a limited tilt range. In conclusion, the claim to being the first is a big too much, as this is certainly not a methodical advance.

We agree with the reviewer comment, and we modified the text accordingly.

l277 the amount of Trolox 1mM seems quite a lot. Is this needed to suppress triplet states?

Correct, based on our own previous works in the lab, and of others in field, the concentration was found to be acceptable ((Cordes et al., 2009, Gust et al., 2014)). However, we have not investigated the effect of Trolox concentration on photoblinking.

l288 What is the physical pixel size of the camera. The model is not stated, but the back projected size is stated as 190 nm with a 100x objective. This information is also not present in the supplement of Boning2021.

We thank the reviewer for spotting this missing detail. We use two Andor iXon 897 cameras with a physical pixel size of 16µm. We work at a magnification of 80x to adjust the pixel size in the sample plane according to the optimum in Mortensen’s formula.

l290-298 Image analysis.This part is vague on the details. Is the code available? I think as the analysis is not standard this needs to be open. The Poisson-weighted Gaussian MLE, what is that? Fitting a Gaussian function with Poisson likelihood? Will not polarisation effects of the restricted dipole play a role already at 0.95NA? (see Stallinga 2010, Optics Express 18:24461 and Engelhardt 2010, NanoLetters 11:209). The drift correction is non-linear, but for the rest it is totally unclear how it works. However, that is quite important as the acquisition time is a bit less than 1h.

The MATLAB code for our image analysis is currently not fully available. Our PSF model is a 2D Gaussian that we fit with a Poisson likelihood. As the reviewer suspects, the dipole orientation does introduce a minor localization bias even at a numerical aperture of 0.95. However, the extent of this error is considerably reduced by selective excitation in epi-illumination and by postselection (at the expense of reducing the yield of the measurement). Furthermore, since the dipole orientation was found to be random with respect to the protein structure, the residual shift is in a random direction and therefore averaged out in the 3D reconstruction.

As for the rest of the image processing pipeline: we correct for drift in a coarse fashion based on image cross-correlation. To this end, we render a super-resolution image every 500 frames (batch size depends on frame rate and SNR) from all localizations. Since each particle has a finite spatial extent with several fluorophores contributing to the localizations, this procedure is not yet sufficient for our purpose. To push the residual drift below 1 nm, we e apply another correction step using occasional single fluorophores as fiducial markers, after they have been distinguished by their polarization, and subtract this averaged drift trajectory. A similar issue arises in image registration: in order to achieve highest accuracy, one needs to identify individual fluorophores as fiducial markers for a non-linear mapping between the imaging channels. Therefore, after coarse registration based on an affine transformation to pair up PSFs, we establish the non-linear matrix for precise registration based on the same single fluorophores identified previously. We have attempted to clarify the workflow in the methods section.

Image analysis

L327” We analyzed raw image stacks from two polarization channels with custom-written MATLAB software. Briefly, we perform dual-channel localization using maximum-likelihood estimation with a Gaussian PSF model. The minor asymmetry due to the 3D dipole orientation causes a small bias of less than 3 nm on average after filtering the data. Mechanical drift during the long acquisition times is first corrected based on image cross-correlation. Localized coordinates from both channels are then registered via an affine transformation and a non-linear correction. Registered PSFs are grouped and their polarization is calculated from the intensities. Single fluorophores are identified via this feature and used as fiducial markers to perform additional drift correction and establish a more precise image alignment via an interpolated map. The residual drift is well below 1 nm and the median registration error is less than 2 nm. Polarization time-traces of multi-fluorophore particles are then analyzed in more detail (see Figure 1 —figure supplement 3) by a routine that is based on the DISC algorithm (White et al., 2020) to find the polarization states of multi-fluorophore conjugated particles. Once all polarization states are identified, the associated coordinates are averaged by taking their median to generate a 2D super-resolved image. We took particles which were fitted best to three polarization states and used them for further analysis (see Table supplement 1 for statistics).”

References

Backer, A. S., Lee, M. Y. and Moerner, W. E. 2016. Enhanced DNA imaging using super-resolution microscopy and simultaneous single-molecule orientation measurements. *Optica,* 3**,** 659.

Böning, D., Wieser, F.-F. and Sandoghdar, V. 2021. Polarization-Encoded Colocalization Microscopy at Cryogenic Temperatures. *ACS Photonics,* 8**,** 194-201.

Cheng, Y., Grigorieff, N., Pawel and Walz, T. 2015. A Primer to Single-Particle Cryo-Electron Microscopy. *Cell,* 161**,** 438-449.

Cordes, T., Vogelsang, J. and Tinnefeld, P. 2009. On the Mechanism of Trolox as Antiblinking and Antibleaching Reagent. *Journal of the American Chemical Society,* 131**,** 5018-5019.

Curd, A. P., Leng, J., Hughes, R. E., Cleasby, A. J., Rogers, B., Trinh, C. H., Baird, M. A., Takagi, Y., Tiede, C., Sieben, C., Manley, S., Schlichthaerle, T., Jungmann, R., Ries, J., Shroff, H. and Peckham, M. 2021. Nanoscale Pattern Extraction from Relative Positions of Sparse 3D Localizations. *Nano Letters,* 21**,** 1213-1220.

Gust, A., Zander, A., Gietl, A., Holzmeister, P., Schulz, S., Lalkens, B., Tinnefeld, P. and Grohmann, D. 2014. A Starting Point for Fluorescence-Based Single-Molecule Measurements in Biomolecular Research. *Molecules,* 19**,** 15824-15865.

Hafi, N., Grunwald, M., Van Den Heuvel, L. S., Aspelmeier, T., Chen, J.-H., Zagrebelsky, M., Schütte, O. M., Steinem, C., Korte, M., Munk, A. and Walla, P. J. 2014. Fluorescence nanoscopy by polarization modulation and polarization angle narrowing. *Nature Methods,* 11**,** 579-584.

Hulleman, C. N., Huisman, M., Moerland, R. J., Grünwald, D., Stallinga, S. and Rieger, B. 2018. Fluorescence Polarization Control for On-Off Switching of Single Molecules at Cryogenic Temperatures. *Small Methods,* 2**,** 1700323.

Hulleman, C. N., Thorsen, R. Ø., Kim, E., Dekker, C., Stallinga, S. And Rieger, B. 2021. Simultaneous orientation and 3D localization microscopy with a Vortex point spread function. *Nature Communications,* 12.

Li, J. K., Wang, N. And Wu, X. S. 1998. Poly(vinyl alcohol) nanoparticles prepared by freezing–thawing process for protein/peptide drug delivery. *Journal of Controlled Release,* 56**,** 117-126.

Lieb, M. A., Zavislan, J. M. and Novotny, L. 2004. Single-molecule orientations determined by direct emission pattern imaging. *Journal of the Optical Society of America B,* 21**,** 1210-1215.

Merk, A., Bartesaghi, A., Banerjee, S., Falconieri, V., Rao, P., Davis, M. I., Pragani, R., Boxer, M. B., Earl, L. A., Milne, J. L. S. and Subramaniam, S. 2016. Breaking Cryo-EM Resolution Barriers to Facilitate Drug Discovery. *Cell,* 165**,** 1698-1707.

Mortensen, K. I., Churchman, L. S., Spudich, J. A. and Flyvbjerg, H. 2010. Optimized localization analysis for single-molecule tracking and super-resolution microscopy. *Nature Methods,* 7**,** 377-381.

Portet, S. 2020. A primer on model selection using the Akaike Information Criterion. *Infectious Disease Modelling,* 5**,** 111-128.

Rosenthal, P. B. and Henderson, R. 2003. Optimal Determination of Particle Orientation, Absolute Hand, and Contrast Loss in Single-particle Electron Cryomicroscopy. *Journal of Molecular Biology,* 333**,** 721-745.

Stallinga, S. and Rieger, B. 2012. Position and orientation estimation of fixed dipole emitters using an effective Hermite point spread function model. *Optics Express,* 20**,** 5896.

White, D. S., Goldschen-Ohm, M. P., Goldsmith, R. H. And Chanda, B. 2020. Top-down machine learning approach for high-throughput single-molecule analysis. *eLife,* 9.

Zhanghao, K., Chen, X., Liu, W., Li, M., Liu, Y., Wang, Y., Luo, S., Wang, X., Shan, C., Xie, H., Gao, J., Chen, X., Jin, D., Li, X., Zhang, Y., Dai, Q. And Xi, P. 2019. Super-resolution imaging of fluorescent dipoles via polarized structured illumination microscopy. *Nature Communications,* 10.

[Editors’ note: further revisions were suggested prior to acceptance, as described below.]

Essential revisions:1) Please, consider answering the 3 specific comments from Reviewer #3, which need no new experiments, just clarification in the text and maybe some simple extra data analysis.

We addressed all reviewer #3 comments below.

Reviewer #3 (Recommendations for the authors):Overall the authors did a very nice job with the revision. I would support the publication of the manuscript.

We thank the reviewer for the insightful and instructive comments again, and we are happy that he/she is satisfied with our previous answers.

point (2) I strongly disagree that classification is needed and an assumption of symmetry. Many published averaging techniques for LM deal exactly with the problem described by the authors "In that sense, the incompletely labelled object inherently behaves as a heterogenous sample with multiple labelling configurations and distances". This is why most of the cryo-EM algorithms fail when applied to strongly under labelled data, they see the different labelling states as different classes. Please have a look at the work of Heydarian et al., (Nat Methods 2018, Nat Com 2021), Sieben et al., (Nat Methods 2018), Salas et al., (PNAS 2017), Shi et al., (PLOS one 2019) and maybe others.The question at hand is, if the under labelling here is so severe that indeed the above methods cannot deliver a good reconstruction. From my own experience somewhere between 30-50% degree of labelling is needed to obtain a meaningful reconstruction. However, this depends also on the absolute number of sites. Some of the above papers have open access code, it might be worthwhile to try this at least.The approach to validate their model via the AIC is good. The idea from Curd et al., have been well incorporated as this method typically only can learn something for the nearest neighbour distance, exactly what it has been used for her.

We thank the reviewer for his/her suggestion. In general, other approaches could be used to deal with the data presented in our study. However, to the best of our knowledge there is no algorithm which is specifically designed for our problem, i.e., a partially labeled, randomly oriented structure with only a few fluorophores and 2D localizations. Nevertheless, we followed the reviewer’s suggestion and tested the code published by Heydarian et al., Nat Methods 2018.

First, we fed the coordinates of our 2D super-resolved *experimental* images obtained from the set of 467 particles used to generate the hexamer structure into the algorithm. As shown in Author response image 3, no clear reconstruction was obtained. To explore this code further, we simulated 3D rotations to the structure “tud_ralf” use in Heydarian et al. Nat Methods 2018 with 100% labelling efficiency. However, we applied rotations around the x axis within ± 180 degrees (i.e., not only in-plane rotations as in the original code) and used the code all2all to solve for the structure. Again, Author response image 4 shows that no clear reconstruction was obtained. Last, we also used a *simulated* data set of our own hexamer structure to avoid all noise. Here, we simulated 500 particles with 900 localization each, at localization precision below 1 nm, labelling efficiency of 50%, and for different 3D orientations in order to mimic our experimental conditions. Again, the algorithm fails to deliver a clear reconstruction (Author response image 5). We also remark that Sieben et al., Nat Methods 2018 do account for the 3D particle orientation, and they are able to separate different species with multi-color imaging. However, their algorithm is based on a cryo-EM classification, which does not deal with strongly under-labelled data. Similarly, the work published by Salas et al., (PNAS 2017).

We believe, there are several factors that limit the use of the other published methods. The first issue concerns the fact that in our samples, structure are randomly oriented in 3D whereas the existing algorithms accounts for translations and rotations in the image plane, i.e., they require particles to be oriented in the top view. Furthermore, our structures are strongly under-labelled and contain classes that are quite similar in terms of sides lengths and angles. It would indeed be interesting to develop a rigorous mathematical formalism for analyzing the performance of each algorithm for structures of different morphological nature.

Recent reports on 3D measurements (Heydarian et al., Nat Com 2021) offer an interesting alternative to reconstruction based on 2D projections. These are currently not applicable to our data as we only have access to 2D measurements. We now mention the prospect of 3D measurements and refer to the general issues in the modified manuscript and Figure 4 —figure supplement 1:

L245: “Classification and reconstruction have recently been applied to other single-molecule localization microscopy studies, and in some cases specific algorithms have been developed to handle some degree of partial labeling (Heydarian et al., 2018, Sieben et al., 2018, Salas et al., 2017). As shown in Figure 4 —figure supplement 1, however, the existing algorithms do not provide a satisfactory solution for our data. We attribute this to the fact that the structures in our investigations are randomly oriented in 3D, are strongly under-labeled, and involve classes that are similar. A mathematical analysis of the performance of various algorithms for structures of different morphologies is beyond the scope of our current work.”

L284: “In addition, while we currently screen different symmetries to find the best model that fits to the structures under study, we plan to explore unsupervised classification schemes for identifying and classifying particles in an unknown sample. In addition, one can exploit the axial information directly and follow similar approach by (Heydarian et al., 2021).”

**Author response image 3. sa2fig3:** Experimental data obtained from partially labelled hexamer protein, as described in the main text. Our experimental data samples 3D different orientations. (**a-b**) Examples of different initial 2D projections. (**c**) We fed the coordinates and localization precision of each of the 467 particles to the all2all algorithm developed by Heydarian et al., (Heydarian et al., 2018). The final reconstruction results fail to deliver the ground truth structural model. Instead, we obtained a scatter of points.

**Author response image 4. sa2fig4:** Simulation of the structural model “tud_ralf” from (Heydarian et al., 2018) at 100% labelling efficiency (see inset in **c**) under the application of 3D rotations (± 180 degrees) around the x-axis only. (**a-b**) Examples of different initial arrangements for two different orientations. (**c**) 256 particles were fed to the all2all algorithm (Heydarian et al., 2018). The final reconstruction fails to deliver the ground truth structural model.

**Author response image 5. sa2fig5:** Simulation data of our hexamer structural model at 50% labeling efficiency. Here we simulated 500 particles with 900 localizations each and localization precision below 1 nm, but sampling 3D different orientations. (**a-b**) Example of different initial particles, adopting different 3D orientations. (**c**) We fed the data to the all2all algorithm (Heydarian et al., 2018), and the final reconstruction, fails to deliver the ground truth structural model.

point (5) The average best position of a set of localisations from the same molecule is typically done by photon weighted averaging not median computation? If there are strong outliers which suggest the use of the median, maybe it would be better to remove the outliers first and then compute a weighted average?

We followed up on the reviewer’s hint at the difference between median and photon-weighted mean in the calculation of the coordinates. Indeed, even in the absence of strong outliers (see scatter plot example, Author response image 6) there is a minor difference in the coordinates, but it is less than 0.3 nm on average. We argue that since this is less than the localization precision, and small compared to the intermolecular distances in our sample, both estimators for the fluorophore position are acceptable. However, we share the reviewer’s view that the photon-weighted mean should be preferred, as it accounts for the different variances of the data points and results in a smaller standard error, especially in cases with higher fluctuation of the obtained photons per frame. For future experiments, we will therefore change the calculation of coordinates to photon-weighted means.

**Author response image 6. sa2fig6:** Scatter plot showing the difference between median and photon-weighted mean in the calculation of the coordinates. Even in the absence of strong outliers (**a**) there is a minor difference (less than 0.3 nm on average) in the coordinates (**b**). Please note the different axis scales between (**a**) and (**b**).

point (13) I see that the author fit a 2D Gaussian, at the expense of a bit of blurring due to model mismatch. I would expect that the random dipole orientation do not average about but give rise to a large standard deviation of the localization than strictly needed with a correct model fitting. The localization will be unbiased due to the randomness, but it will result in a bit of extra blurring.

The reviewer is right and we would like to clarify this point again. The localizations contain a small error due to the model mismatch of a Gaussian with a dipole-like PSF. This However, in order to obtain the final reconstruction, we collect many particles. Since the dipole orientation is random, the small shift occurs in a random direction. The ‘averaged’ 3D structure will therefore be unbiased. We have answered a similar question by reviewer 1:

“The reviewer points to an important consideration. The localization bias due to the asymmetry of the dipole PSF is a major contribution to the overall localization uncertainty. However, the overall extent of this contribution is reduced by the fact that we preferentially excite the in-plane components and detect them more efficiently due to the lower NA. In addition, by discriminating against less bright fluorescence signals, we eliminate contributions from molecules that have dipole moments with substantial axial component. We estimate the contribution to the localization inaccuracy to be no more than 3 nm on average as was discussed in our recent publication (Böning et al., ACS Photonics 2021). Nevertheless, it is possible to reach sub-nanometer resolution with a more selective choice of the particles that are included in the averaging process, as validated in Figure 2b-d and Figure 3b”

References

Heydarian, H., Joosten, M., Przybylski, A., Schueder, F., Jungmann, R., Werkhoven, B. V., Keller-Findeisen, J., Ries, J., Stallinga, S., Bates, M. and Rieger, B. 2021. 3D particle averaging and detection of macromolecular symmetry in localization microscopy. *Nature Communications,* 12.

Heydarian, H., Schueder, F., Strauss, M. T., Van Werkhoven, B., Fazel, M., Lidke, K. A., Jungmann, R., Stallinga, S. and Rieger, B. 2018. Template-free 2D particle fusion in localization microscopy. *Nature Methods,* 15**,** 781-784.

Salas, D., Le Gall, A., Fiche, J.-B., Valeri, A., Ke, Y., Bron, P., Bellot, G. and Nollmann, M. 2017. Angular reconstitution-based 3D reconstructions of nanomolecular structures from superresolution light-microscopy images. *Proceedings of the National Academy of Sciences,* 114**,** 9273-9278.

Sieben, C., Banterle, N., Douglass, K. M., Gönczy, P. and Manley, S. 2018. Multicolor single-particle reconstruction of protein complexes. *Nature Methods,* 15**,** 777-780.